# PRIVACY PRESERVING RECALIBRATION UNDER DOMAIN SHIFT

## ABSTRACT

Classifiers deployed in high-stakes applications must output calibrated confidence scores, i.e. their predicted probabilities should reflect empirical frequencies. Typically this is achieved with recalibration algorithms that adjust probability estimates based on real-world data; however, existing algorithms are not applicable in real-world situations where the test data follows a different distribution from the training data, and privacy preservation is paramount (e.g. protecting patient records). We introduce a framework that provides abstractions for performing recalibration under differential privacy constraints. This framework allows us to adapt existing recalibration algorithms to satisfy differential privacy while remaining effective for domain-shift situations. Guided by our framework, we also design a novel recalibration algorithm, accuracy temperature scaling, that is tailored to the requirements of differential privacy. In an extensive empirical study, we find that our algorithm improves calibration on domain-shift benchmarks under the constraints of differential privacy. On the 15 highest severity perturbations of the ImageNet-C dataset, our method achieves a median ECE of 0.029, over 2x better than the next best recalibration method and almost 5x better than without recalibration.

## 1 INTRODUCTION

Machine learning classifiers are currently deployed in high stakes applications where (1) the cost of failure is high, so prediction uncertainty must be accurately calibrated (2) the test distribution does not match the training distribution, and (3) data is subject to privacy constraints. All three of these challenges must be addressed in applications such as medical diagnosis (Khan et al., 2001; Chen et al., 2018; Kortum et al., 2018), financial decision making (Berestycki et al., 2002; Rasekhschaffe & Jones, 2019; He & Antón, 2003), security and surveillance systems (Sun et al., 2015; Patel et al., 2015; Agre, 1994), criminal justice (Berk, 2012; 2019; Rudin & Ustun, 2018), and mass market autonomous driving (Kendall & Gal, 2017; Yang et al., 2018; Glancy, 2012). While much prior work has addressed these challenges individually, they have not been considered simultaneously. The goal of this paper is to propose a framework that formalizes challenges (1)-(3) jointly, introduce benchmark problems, and design and compare new algorithms under the framework.

A standard approach for addressing challenge (1) is uncertainty quantification, where the classifier outputs its confidence in every prediction to indicate how likely it is that the prediction is correct. These confidence scores must be meaningful and trustworthy. A widely used criterion for good confidence scores is calibration (Brier, 1950; Cesa-Bianchi & Lugosi, 2006; Guo et al., 2017) — i.e. among the data samples for which the classifier outputs confidence $p \in (0, 1)$, exactly $p$ fraction of the samples should be classified correctly.

Several methods (Guo et al., 2017) learn calibrated classifiers when the training distribution matches the test distribution. However, this classical assumption is always violated in real world applications, and calibration performance can significantly degrade under even small domain shifts (Snoek et al., 2019). To address this challenge, several methods have been proposed to re-calibrate a classifier on data from the test distribution (Platt et al., 1999; Guo et al., 2017; Kuleshov et al., 2018; Snoek et al., 2019). These methods make small adjustments to the classifier to minimize calibration error on a validation dataset drawn from the test distribution, but they are typically only applicable when they have (unrestricted) access to data from this validation set.

Additionally, high stakes applications often require privacy. For example, it is difficult for hospitals to share patient data with machine learning providers due to legal privacy protections (Centers for

Medicare & Medicaid Services, 1996). When the data is particularly sensitive, provable differential privacy becomes necessary. Differential privacy (Dwork et al., 2014) provides a mathematically rigorous definition of privacy along with algorithms that meet the requirements of this definition. For instance, the hospital may share only certain statistics of their data, where the shared statistics must have bounded mutual information with respect to individual patients. The machine learning provider can then use these shared statistics — possibly combining statistics from many different hospitals — to recalibrate the classifier and provide better confidence estimates.

In this paper, we present a framework to address all three challenges – calibration, domain shift, and differential privacy – and introduce a benchmark to standardize performance and compare algorithms. We show how to modify modern recalibration techniques (e.g. (Zadrozny & Elkan, 2001; Guo et al., 2017)) to satisfy differential privacy using this framework, and compare their empirical performance. This framework can be viewed as performing federated learning for recalibration, with the constraint that each party's data must be kept differentially private.

We also present a novel recalibration technique, accuracy temperature scaling, that is particularly effective in this framework. This new technique requires private data sources to share only two statistics: the overall accuracy and the average confidence score for a classifier. We adjust the classifier until the average confidence equals the overall accuracy. Because only two numbers are revealed by each private data source, it is much easier to satisfy differential privacy. In our experiments, we find that without privacy requirements the new recalibration algorithm performs on par with algorithms that use the entire validation dataset, such as (Guo et al., 2017); with privacy requirements the new algorithm performs 2x better than the second best baseline.

In summary, the contributions of our paper are as follows. (1) We introduce the problem of "privacy preserving calibration under domain shift" and design a framework for adapting existing recalibration techniques to this setting. (2) We introduce accuracy temperature scaling, a novel recalibration method designed with privacy concerns in mind, that requires only the overall accuracy and average confidence of the model on the validation set. (3) Using our framework, we empirically evaluate our method on a large set of benchmarks against state-of-the-art techniques and show that it performs well across a wide range of situations under differential privacy.

## 2 BACKGROUND AND RELATED WORK

### 2.1 CALIBRATION

**Description of Calibration** Consider a classification task from input domain (e.g. images) $\mathcal{X}$ to a finite set of labels $\mathcal{Y} = \{1, \cdots, m\}$. We assume that there is some joint distribution $P^*$ on $\mathcal{X} \times \mathcal{Y}$. This could be the training distribution, or the distribution from which we draw test data. A classifier is a pair $(\phi, \hat{p})$ where $\phi : \mathcal{X} \to \mathcal{Y}$ maps each input $x \in \mathcal{X}$ to a label $y \in \mathcal{Y}$ and $\hat{p} : \mathcal{X} \to [0, 1]$ maps each input $x$ to a confidence value $c$. We say that the classifier $(\phi, \hat{p})$ is perfectly calibrated (Brier, 1950; Gneiting et al., 2007) with respect to the distribution $P^*$ if $\forall c \in [0, 1]$

$$\Pr_{P^*(x,y)}[\phi(x) = y \mid \hat{p}(x) = c] = c. \tag{1}$$

Note that calibration is a property not only of the classifier $(\phi, \hat{p})$, but also of the distribution $P^*$. A classifier $(\phi, \hat{p})$ can be calibrated with respect to one distribution (e.g. the training distribution) but not another (e.g. the test distribution). To simplify notation we drop the dependency on $P^*$.

To numerically measure how well a classifier is calibrated, the commonly used metric is Expected Calibration Error (ECE) (Naeini et al., 2015) defined by

$$\text{ECE}(\phi, \hat{p}) := \int_{c \in [0,1]} \Pr[\hat{p}(x) = c] \cdot |\Pr[\phi(x) = y \mid \hat{p}(x) = c] - c| \,. \tag{2}$$

In other words, ECE measures average deviation from Eq. 1. In practice, the ECE is approximated by binning — partitioning the predicted confidences into bins, and then taking a weighted average of the difference between the accuracy and average confidence for each bin (see Appendix A.1 for details.)

**Recalibration Methods** Several methods apply a post-training adjustment to a classifier $(\phi, \hat{p})$ to achieve calibration (Platt et al., 1999; Niculescu-Mizil & Caruana, 2005). The one most relevant to our paper is temperature scaling (Guo et al., 2017). On each input $x \in \mathcal{X}$, a neural network

typically first computes a logit score $l_1(x), l_2(x), \cdots, l_n(x)$ for each of the $n$ labels, then computes a confidence score or probability estimate $\hat{p}(x)$ with a softmax function. Temperature scaling adds a temperature parameter $T \in \mathbb{R}^+$ to the softmax function

$$\hat{p}(x; T) = \max_i \frac{e^{l_i(x)/T}}{\sum_j e^{l_j(x)/T}}. \qquad (3)$$

A higher temperature reduces the confidence, and vice versa. $T$ is trained to minimize the standard cross entropy objective on the validation dataset, which is equivalent to maximizing log likelihood. Despite its simplicity, temperature scaling performs well empirically in classification calibration for deep neural networks.

Alternative methods for classification calibration have also been proposed. Histogram binning (Zadrozny & Elkan, 2001) partitions confidence scores $\in [0, 1]$ into bins $\{[0, \epsilon), [\epsilon, 2\epsilon), \cdots, [1 - \epsilon, 1]\}$ and sorts each validation sample into a bin based on its confidence $\hat{p}(x)$. The algorithm then resets the confidence level of each bin to match the average classification accuracy of data points in that bin. Isotonic regression methods (Kuleshov et al., 2018) learn an additional layer on top of the softmax output layer. This additional layer is trained on a validation dataset to fit the output confidence scores to the empirical probabilities in each bin. Other methods include Platt scaling (Platt et al., 1999) and Gaussian process calibration (Wenger et al., 2019).

## 2.2 ROBUSTNESS TO DOMAIN SHIFT

Preventing massive performance degradation of machine learning models under domain shift has been a long-standing problem. There are several approaches developed in the literature. Unsupervised domain adaptation (Ganin & Lempitsky, 2014; Shu et al., 2018) learns a joint representation between the source domain (original data) and target domain (domain shifted data). Invariance based methods (Cissé et al., 2017; Miyato et al., 2018; Madry et al., 2017; Lakshminarayanan et al., 2017; Cohen et al., 2019) prevent the classifier output from changing significantly given small perturbations to the input. Transfer learning methods (Pan & Yang, 2009; Bengio, 2012; Dai et al., 2007) fine-tune the classifier on labeled data in the target domain. We classify our method in this category because we also fine-tune on the target domain, but with minimal data requirements (we only need the overall classifier accuracy).

## 2.3 DIFFERENTIAL PRIVACY

Differential privacy (Dwork et al., 2014) is a procedure for sharing information about a dataset to the public while withholding critical information about individuals in the dataset. Informally, it guarantees that an attacker can only learn a limited amount of new information about an individual. Differentially private approaches are critical in privacy sensitive applications. For example, a hospital may wish to gain medical insight or calibrate its prediction models by releasing diagnostic information to outside experts, but it cannot release information about any particular patient.

One common notion of differential privacy is $\epsilon$-differential privacy (Dwork et al., 2014). Let us define a database $D$ as a collection of data points in a universe $\mathcal{X}$, and represent it by its histogram: $D \in \mathbb{N}^{|\mathcal{X}|}$, where each entry $D_x$ represents the number of elements in the database that takes the value $x \in \mathcal{X}$. A randomized algorithm $\mathcal{M}$ is one that takes in input $D \in \mathbb{N}^{|\mathcal{X}|}$ and (stochastically) outputs some value $\mathcal{M}(D) = b$ for $b \in \text{Range}(\mathcal{M})$.

**Definition 1.** *Let $\mathcal{M}$ be a randomized function $\mathcal{M} : \mathbb{N}^{|\mathcal{X}|} \to \text{Range}(\mathcal{M})$. We say that $\mathcal{M}$ is $\epsilon$-differentially private if for all $\mathcal{S} \subseteq \text{Range}(\mathcal{M})$ and for any two databases $D, D' \in \mathbb{N}^{|\mathcal{X}|}$ that differ by only one element, i.e. $\|D - D'\|_1 \leq 1$, we have*

$$\frac{\Pr[\mathcal{M}(D) \in \mathcal{S}]}{\Pr[\mathcal{M}(D') \in \mathcal{S}]} \leq e^{\epsilon}$$

Intuitively, the output of $\mathcal{M}$ should not change much if a single data point is added or removed. An attacker that learns the output of $\mathcal{M}$ gains only limited information about any particular data point.

Given a deterministic real valued function $f : \mathbb{N}^{|\mathcal{X}|} \to \mathbb{R}^z$, we would like to design a function $\mathcal{M}$ that remains as close as possible to $f$ but satisfies Definition 1. This can be achieved by the Laplace

mechanism (McSherry & Talwar, 2007; Dwork, 2008). Let us define the $L_1$ sensitivity of $f$:

$$\Delta f = \max_{\substack{D,D' \in \mathbb{N}^{|\mathcal{X}|} \\ \|D-D'\|_1 = 1}} \|f(D) - f(D')\|_1$$

Then the Laplace mechanism adds Laplacian random noise as in (4):

$$\mathcal{M}_L(D; f, \epsilon) = f(D) + (Y_1, \ldots, Y_z) \tag{4}$$

where $Y_i$ are i.i.d. random variables drawn from the $\text{Laplace}(\text{loc} = 0, \text{scale} = \Delta f/\epsilon)$ distribution. The function $\mathcal{M}_L$ satisfies $\epsilon$-differential privacy, and we reproduce the proof in Appendix A.2.

## 3 RECALIBRATION UNDER DIFFERENTIAL PRIVACY

In this section we propose a framework for performing recalibration that allows independent parties to pool their data for improved calibration, while maintaining differential privacy. This setup can be framed as differentially private federated learning for recalibration. Multiple parties experience the same domain shift (e.g. because they live in the same changing world). Each party would benefit from access to additional data, but each party also wants to keep their own data private. Our framework allows all parties to react to domain shifts more quickly by pooling their data (so each individual party needs less labeled data from the new distribution), while maintaining the privacy of each party.

### 3.1 EXAMPLE APPLICATIONS

We begin with example scenarios that illustrate the main desiderata and challenges of this problem.

**Example 1:** Suppose you have a classifier for diagnosing a medical condition and deploy your classifier across many hospitals. The hospitals need calibrated confidences for a similar but more unusual condition (e.g. the original model may have been trained on an already existing virus strain but need to be recalibrated for a novel strain of the virus). There are two options: 1. Each hospital uses only their own private data to calibrate the classifier; 2. Each hospital sends some (differentially private) information to you, and you aggregate the information and calibrate the classifier. Option 2 is preferable if each hospital has only a handful of patients for the particular condition.

In this case, the hospitals are the parties that wish to keep their data (patient info) private. The novel strain of the virus represents a domain shift. If the hospitals each have only a few data points, they want to aggregate their data in order to improve their classifier's calibration while still respecting patient privacy.

**Example 2:** Suppose that there is a third-party advertising company that runs ads for websites. This advertising company has worked with news websites before, but recently acquired new clients from a different category of websites. The individual websites have user information, but they cannot provide the third-party advertising company with this user information due to privacy constraints. The third-party advertising company wants calibrated models for whether a user will click on an ad.

**Example 3:** Another category of scenarios in which our framework can be used is for individual privacy. An individual may have labeled data that he wishes to keep private; however, he would still like calibrated confidences from prediction models (e.g. financial software for individuals). With differential privacy, individuals can provide summary statistics with added noise to an aggregator. In this setup, differential privacy is guaranteed on the individual level. Aggregators can then improve their confidence estimation using noisy summary statistics from many individuals.

### 3.2 GENERAL FRAMEWORK

We propose a standard framework to handle the general situation represented by the examples above. This two-party framework involves (1) a calibrator and (2) private data sources, and it allows us to adapt recalibration algorithms for differential privacy. A private data source may be e.g. a hospital (as in Example 1 above), a website (as in Example 2), or an individual (as in Example 3).

1. [Calibrator:] Input an uncalibrated classifier $(\phi, \hat{p})$.
2. [Private Data Sources:] Each data source $i = 1, \cdots, d$ inputs private dataset $D_i$.

3. At iteration $k = 1, \cdots, K$

   (a) [Calibrator:] The calibrator designs a function $f^k : \mathbb{N}^{|\mathcal{X}|} \to \mathbb{R}^s$, where $s \in \mathbb{N}$. For each $i = 1, \cdots, d$, the calibrator sends function $f^k$ to private data source $i$.

   (b) [Private Data Sources:] For each $i = 1, \cdots, d$, the $i$-th private data source uses the Laplace mechanism in Eq. 4 to convert $f^k$ to $\mathcal{M}^k$ that satisfies $\epsilon/K$-differential privacy, and sends $\mathcal{M}^k(D_i)$ back to the calibrator.

4. [Calibrator:] Output a new classifier $(\phi, \hat{p}')$ based on $\mathcal{M}^k(D_i), k = 1, \cdots, K, i = 1, \cdots, d$.

Under this framework, differential privacy is automatically satisfied: if for each $k = 1, \cdots, K$, $\mathcal{M}^k$ is $\epsilon/K$-differentially private, then the combined function $(\mathcal{M}^1, \cdots, \mathcal{M}^k)$ is $\epsilon$-differentially private (Dwork et al., 2014). The differential privacy guarantees for each private data source $i$ are independent of the policy of the calibrator or other private data sources; i.e. even if the calibrator and all other private data sources collude to steal information from the $i$-th data source — as long as the $i$-th private data source follows the protocol, its data will be protected by differential privacy.

This framework simplifies the problem into two design choices: select the query function $f^k$ for $k = 1, \cdots, K$, and select the mapping from observations $\mathcal{M}^k(D_1), \cdots, \mathcal{M}^k(D_d)$ at $k = 1, \cdots, K$ to the calibrated confidence function $\hat{p}'$. We will discuss the most reasonable choices for several existing recalibration algorithms. Note that in general, the calibration quality degrades as the privacy level increases (i.e. $\epsilon$ decreases).

### 3.3 Adapting Existing Algorithms

In this section, we explain how we adapt algorithms introduced in Section 2 to our framework. Note that many existing recalibration algorithms involve parametric optimization, and in these cases multiple iterations $K$ are needed to search the parameter space. However, using additional iterations hurts the calibration since a larger $K$ increases the added Laplace noise for $\epsilon/K$-differential privacy; i.e. for any fixed Laplace noise, more queries means less privacy. Thus, we propose the use of the golden section search algorithm as a better alternative to grid search for parametric optimization, since it is more efficient at finding the extremum of a unimodal function within a specified interval, and requires fewer queries. See Appendix C.1 for additional details about the golden section search.

**Temperature Scaling** Temperature scaling finds the temperature $T$ in Eq. 3 that maximizes log likelihood. At each iteration $k = 1, \cdots, K$, the function $f^k$ queries $D_i$ for the log likelihood at some temperature, and we average the log likelihood over all the private datasets. We observe that log likelihood is a unimodal function of the temperature in Proposition 1 (see Appendix B for proof). Therefore, the golden section search algorithm can find the maximum of the unimodal function with the fewest queries. We may refer to temperature scaling as NLL-T for brevity.

**Proposition 1.** *For any distribution $p^*$ on $\mathcal{X} \times \mathcal{Y}$ where $\mathcal{Y} = \{1, \cdots, m\}$, and for any set of functions $l_1, \cdots, l_m : \mathcal{X} \to \mathbb{R}$, $\mathbb{E}_{x,y \sim p^*} \left[ \log \frac{e^{l_y(x)/T}}{\sum_j e^{l_j(x)/T}} \right]$ is a unimodal function of $T$.*

**ECE Minimization (ECE-T)** Instead of finding a temperature that maximizes log likelihood, we find that empirically it is often better to directly minimize the discretized ECE in Eq. 2. Adapting ECE minimization to our framework is similar to log likelihood maximization, except that we query for the necessary quantities to compute the ECE score instead of the log likelihood. In Appendix C.2.2, we show how to compute the ECE score with as few queried quantities as possible.

**Histogram Binning** Histogram binning can be adapted to the above protocol with only one iteration ($K = 1$). The function $f^1$ queries $D_i$ for the number of correct predictions in each bin and the total number of samples in each bin. We average the query results from different datasets. To compute the new confidence for a bin, we divide the average number of correct predictions by the average total number of samples in each bin.

## 4 Accuracy Temperature Scaling

When we add Laplace noise according to Eq. 4, the added noise increases with the number of iterations $K$ and the $L_1$ sensitivity of the query functions $f^k$. In other words, when we adapt a calibration algorithm to our framework, we need to add more noise if the original algorithm gains a

lot of information about the private datasets $D_1, \cdots, D_d$. The relative amount of noise also increases as the amount of data available decreases, as is the case when binning is used. Larger noise will degrade calibration performance. To improve performance, we propose a new recalibration algorithm called accuracy temperature scaling that acquires much less information than previous algorithms.

Our method is a form of temperature scaling that is based on a weaker notion than calibration. Let classification accuracy and average confidence be denoted as

$$\text{Acc}(\phi) = \Pr[\phi(x) = y], \text{ and } \text{Conf}(\hat{p}) = \mathbb{E}[\hat{p}(x)]$$

Acc and Conf are expectations of $[0, 1]$-bounded random variables, so they can be accurately estimated even from a relatively small quantity of data. We say that a classifier is consistent if $\text{Acc}(\phi) = \text{Conf}(\hat{p})$. We tune the temperature parameter in Eq. 3 until the average confidence Conf is identical to the average accuracy Acc, i.e. until consistency is achieved. We will refer to our method as Acc-T for brevity.

Consistency is a strictly weaker condition than calibration. Surprisingly, even when there is a lot of data and no privacy requirements, optimizing for consistency achieves similar performance as directly optimizing for ECE in our experiments, as shown in Appendix E.2.

### 4.1 ACCURACY TEMPERATURE SCALING UNDER DIFFERENTIAL PRIVACY

Adapting Acc-T to our differential privacy framework is similar to doing so for temperature scaling in Section 3.3. As we show in Proposition 2 (see Appendix B for proof), the Acc-T objective is also a unimodal function of $T$, so we can use golden section search to find the $T$ that minimizes the objective function. Algorithm 1 provides the complete algorithm for Acc-T under differential privacy. On Line 2, we select initial temperature values. Line 3 specifies a query function that the hospitals use to pool their data while respecting differential privacy. Lines 4-12 implement differentially private golden section search over the recalibration temperature parameter. The algorithm outputs a temperature value that improves the classifier's calibration on the new domain.

**Proposition 2.** *For any distribution $p^*$ on $\mathcal{X} \times \mathcal{Y}$ where $\mathcal{Y} = \{1, \cdots, m\}$, and for any set of functions $l_1, \cdots, l_m : \mathcal{X} \to \mathbb{R}$, let $\hat{p}_T : x \mapsto \max_i \frac{e^{l_i(x)/T}}{\sum_j e^{l_j(x)/T}}$ and $\phi : x \mapsto \arg\max_i l_i(x)$. $|\Pr_{x,y \sim p^*}[\phi(x) = y] - \mathbb{E}_{p^*}[\hat{p}_T(x)]|$ is a unimodal function of $T$.*

---

**Algorithm 1** Acc-T with differential privacy

1: **Input** Private datasets $D_1, \cdots, D_d$. Logit functions $l_1, \cdots, l_m : \mathcal{X} \to \mathbb{R}$. Initial temperature range $[T_-^0, T_+^0]$. Number of iterations $K$. Define $\phi$ and $\hat{p}_T$ as in Proposition 2.
2: Set $T_0^0 = T_+^0 - (T_+^0 - T_-^0) * 0.618, T_1^0 = T_-^0 + (T_+^0 - T_-^0) * 0.618$
3: For $T_0^0$ set $\mathcal{M}^0 : D_i \mapsto \sum_{x_i, y_i \in D_i} \left( \mathbb{I}(\phi(x_i) = y_i) - \hat{p}_{T_0^0}(x_i) \right) + \text{Lap}\left(\frac{K+1}{\epsilon}\right)$ and sample $v_0^0 = \frac{1}{d} \sum_{i=1}^d \mathcal{M}^0(D_i)$. Similarly set $\mathcal{M}^1$ for $T_1^0$ and sample $v_1^0$.
4: **for** $k = 0, \cdots, K - 1$ **do**
5:    **if** $|v_0^k| \geq |v_1^k|$ **then**
6:       Set $T_+^{k+1} = T_+^k, T_-^{k+1} = T_0^k, T_0^{k+1} = T_1^k, T_1^{k+1} = T_- + (T_+ - T_-) * 0.618$
7:       Set $v_0^{k+1} = v_1^k$. Sample $v_1^{k+1}$ for $T_1^{k+1}$ as in line 3.
8:    **else**
9:       Set $T_-^{k+1} = T_-^k, T_+^{k+1} = T_1^k, T_1^{k+1} = T_0^k, T_0^{k+1} = T_+ - (T_+ - T_-) * 0.618$
10:      Set $v_1^{k+1} = v_0^k$. Sample $v_0^{k+1}$ for $T_0^{k+1}$ as in line 3.
11:    **end if**
12: **end for**
13: Return $(T_-^K + T_+^K)/2$ as the optimal temperature.

---

### 4.2 COMPARISON

We will briefly discuss how our method, Acc-T, compares to others such as histogram binning, temperature scaling, or ECE-T in terms of its theoretical bias (calibration error given infinite data), worst case variance (calibration error degradation when less data is available), and adaptability to differential privacy (based on the relative amount of noise that must be added to satisfy differential

privacy). Acc-T has a higher theoretical bias than the other methods, since its objective function does not directly minimize the calibration error. However, in our experiments on deep neural networks, the bias of Acc-T is only slightly worse or comparable to that of ECE-T or temperature scaling in practice. Acc-T also has a lower worst case variance than other methods because it does not use binning (so there are more data points per bin) and its objective function has a smaller range than that of temperature scaling. Overall, Acc-T has the highest adaptability to differential privacy; it has smaller $L_1$ sensitivity than the other methods, so less noise is necessary for differential privacy. Additional factors that affect the calibration quality and the level of privacy are discussed in Appendix D.1.

## 5 EXPERIMENTS

In this section, we run an extensive series of large, controllable experiments on three datasets to compare our proposed method Acc-T against five different baseline methods, three of which are designed with privacy concerns in mind, using the general procedure in Section 3. These benchmarks include various domain shifts and privacy settings, and our proposed Acc-T method consistently outperforms the other baseline methods. We also extensively validate the relationship between calibration error and several relevant factors for domain shift and privacy. Additional experimental details are included in Apppendix E.

### 5.1 EXPERIMENTAL SETUP

**Methods** We evaluate the differentially private versions of temperature scaling, ECE-T, histogram binning, and Acc-T over an extensive range of settings that considers calibration under various domain shifts and privacy concerns. We also include two baseline methods, (1) no calibration and (2) recalibration with only one private dataset from the target domain (so data from other sources is not used; in this case privacy constraints need not be taken into account but less data is available).

**Datasets** To simulate various domain shifts, we use the ImageNet-C, CIFAR-100-C, and CIFAR-10-C datasets (Hendrycks & Dietterich, 2019), which are perturbed versions of the ImageNet (Deng et al., 2009), CIFAR-100 (Krizhevsky & Hinton, 2009), and CIFAR-10 (Krizhevsky & Hinton, 2009) test sets. Each -C dataset includes 15 perturbed versions of the original test set, with perturbations such as Gaussian noise, motion blur, jpeg compression, and fog. We divide each perturbed test set into a validation split containing different "private data sources" with the same number of samples, and a test split containing all of the remaining images. We then apply the recalibration algorithms over the validation split and evaluate the ECE on the test split. Note that only the unperturbed training sets were used to train the models.

**Relevant factors** We evaluate the ECE for all of the methods while controlling the following three factors: (1) the number of private data sources, (2) the number of samples per data source, and (3) the privacy level $\epsilon$. When we vary one factor, we keep the other two factors constant.

**Additional details** We use $K = 5$ iterations for all experiments, and report the average ECE achieved over 500 trials with randomly divided splits for each experiment. We report other experimental setup details including the type of network used in Appendix E.1.

### 5.2 RESULTS AND ANALYSIS

In Fig. 1, we plot the ECE vs. (1a) the number of private data sources, (1b) the number of samples per data source, and (1c) the $\epsilon$ value, for the ImageNet "fog" perturbation. Fig. 2 shows a similar plot for the CIFAR-100 "jpeg compression" perturbation, and Fig. 3 shows a similar plot for the CIFAR-10 "motion blur" perturbation. Our proposed method, Acc-T, is shown in red, and clearly outperforms other methods under the constraints of differential privacy for these ranges of values. Full plots for all perturbations and datasets are included in Appendix E.3. Table 1 shows the overall median and mean ECE achieved by each recalibration method on ImageNet, CIFAR-100, and CIFAR-10. These averages are computed over all perturbations, numbers of private data sources, numbers of samples per source, and $\epsilon$ settings from the suite of experiments in E.3. Our method, Acc-T, far outperforms other methods in the domain-shifted differential privacy setting.

The performance of all recalibration algorithms degrades when subjected to the constraints of differential privacy, but some are affected more than others for a given situation. Selecting a

differentially private recalibration algorithm for a particular situation thus requires some consideration. To this end, we provide some analysis over these methods under the three relevant factors.

**Number of Private Data Sources** As the number of sources increases, Acc-T tends to do well, even when the number of samples per source is small. Because Acc-T does not involve binning and the sensitivity of its objective function is small, there is relatively less noise for this method than for others. Therefore, it can effectively combine data from multiple sources even under the constraints of differential privacy, and is the best method in general.

**Number of Samples Per Source** As the number of samples per source increases, Acc-T tends to do well given enough data sources. As the number of samples per source grows towards infinity, recalibration with only one source works very well since we do not need to query other sources or apply privacy constraints. Histogram binning and ECE-T may also perform quite well with many bins when the number of samples is very large.

**Privacy concern $\epsilon$** When $\epsilon$ is very low (i.e. the privacy requirements are very high), recalibrating with only one data source works well; this method remains unaffected by the strong privacy constraints, while all other methods worsen drastically due to the increased noise. For mid-range $\epsilon$ values, Acc-T works well. When $\epsilon$ is very high, ECE-T can work well, since privacy is not much of a concern.

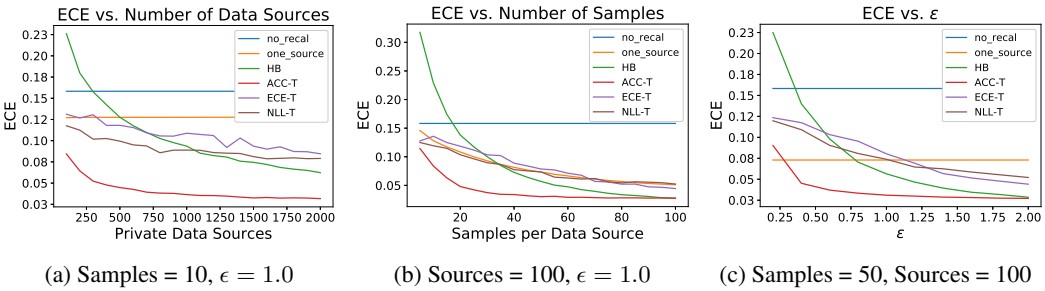

(a) Samples = 10, $\epsilon = 1.0$     (b) Sources = 100, $\epsilon = 1.0$     (c) Samples = 50, Sources = 100

Figure 1: Recalibration results for ImageNet under the "fog" perturbation, with varying (1a) number of private data sources, (1b) number of samples per data source, and (1c) privacy level $\epsilon$. Acc-T does best in these settings.

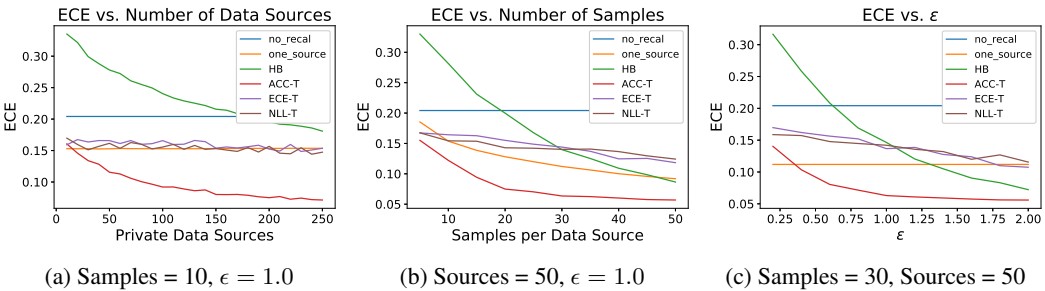

(a) Samples = 10, $\epsilon = 1.0$     (b) Sources = 50, $\epsilon = 1.0$     (c) Samples = 30, Sources = 50

Figure 2: Recalibration results for CIFAR-100 under the "jpeg compression" perturbation, with varying (2a) number of private data sources, (2b) number of samples per source, and (2c) privacy level $\epsilon$. Acc-T does best.

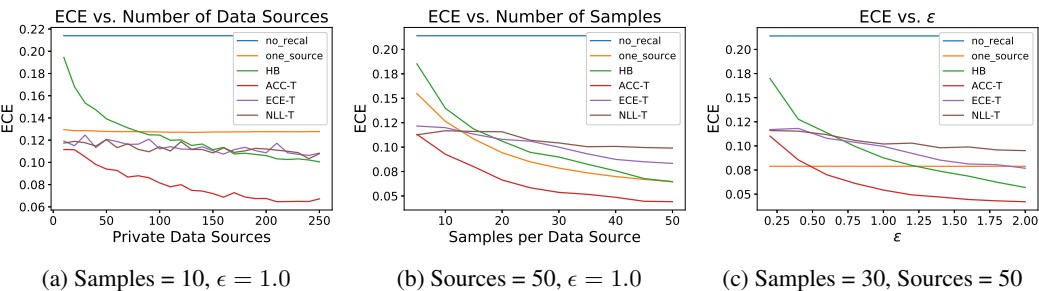

(a) Samples = 10, $\epsilon = 1.0$     (b) Sources = 50, $\epsilon = 1.0$     (c) Samples = 30, Sources = 50

Figure 3: Recalibration results for CIFAR-10 under the "motion blur" perturbation, with varying (3a) number of private data sources, (3b) number of samples per source, and (3c) privacy level $\epsilon$. Acc-T does best.

| Expected Calibration Error (median / mean) | | | |
|---|---|---|---|
| Recalibration method | ImageNet | CIFAR-100 | CIFAR-10 |
| No recalibration | 0.1343 / 0.1334 | 0.2718 / 0.3124 | 0.2187 / 0.2924 |
| One source | 0.0657 / 0.0700 | 0.1204 / 0.1209 | 0.1241 / 0.1359 |
| Histogram binning | 0.0656 / 0.0787 | 0.1867 / 0.1850 | 0.1168 / **0.1181** |
| ECE-T | 0.0684 / 0.0739 | 0.1655 / 0.1721 | 0.1160 / 0.1668 |
| NLL-T | 0.0597 / 0.0624 | 0.1583 / 0.1607 | 0.1157 / 0.1653 |
| Acc-T | **0.0289 / 0.0325** | **0.0890 / 0.0973** | **0.0836** / 0.1199 |

Table 1: Median and mean expected calibration error (ECE) achieved for domain-shifted data under differential privacy. Columns from left to right show the median/mean ECE achieved over all perturbations, number of private data sources, number of samples per source, and $\epsilon$ for ImageNet, CIFAR-100, and CIFAR-10. Best calibration is shown in **bold**.

## 6 CONCLUSION

Simultaneously addressing the challenges of calibration, domain shift, and privacy is extremely important in many environments. In this paper, we introduced a framework for recalibration on domain-shifted data under the constraints of differential privacy. Within this framework, we designed a novel algorithm to handle all three challenges. Our method demonstrated impressive performance across a wide range of settings on a large suite of benchmarks. In future work, we are interested in investigating recalibration under different types of privacy mechanisms.

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

# A ADDITIONAL BACKGROUND INFORMATION

## A.1 COMPUTATION OF ECE

To compute the ECE, discretization is necessary. We first divide $[0, 1]$ into bins $\mathbf{c} = (\mathbf{c}_1, \cdots, \mathbf{c}_k)$ such that $0 < \mathbf{c}_1 < \cdots < \mathbf{c}_k = 1$, and then we compute the average accuracy Acc and average confidence Conf in each bin (for convenience, denote $\mathbf{c}_0 = 0$)

$$\mathrm{Acc}(f, \mathbf{c}, i) = \Pr\left[f(x) = y | \hat{p}(x) \in [\mathbf{c}_{i-1}, \mathbf{c}_i)\right]$$
$$\mathrm{Conf}(f, \mathbf{c}, i) = \mathbb{E}[\hat{p}(x) | \hat{p}(x) \in [\mathbf{c}_{i-1}, \mathbf{c}_i)]$$

Then the ECE defined in Eq. 2 can be approximated by a discretized version

$$\mathrm{ECE}(f, \hat{p}) \approx \mathrm{ECE}(f, \hat{p}; \mathbf{c})$$

$$:= \sum_{i=1}^{k} \Pr\left[\hat{p}(x) \in [\mathbf{c}_{i-1}, \mathbf{c}_i)\right] \cdot |\mathrm{Acc}(f, \mathbf{c}, i) - \mathrm{Conf}(f, \mathbf{c}, i)|$$

Given empirical data $\mathcal{D} = \{x_{1:n}, y_{1:n}\}$ we can estimate $\mathrm{ECE}(f, \hat{p}; \mathbf{c})$ as

$$\mathrm{ECE}(f, \hat{p}; \mathbf{c}) \approx \mathrm{E\hat{C}E}(f, \hat{p}; \mathbf{c}, \mathcal{D})$$

$$:= \sum_{i=1}^{k} \frac{1}{n} \left| \sum_{x_i \in [\mathbf{c}_{i-1}, \mathbf{c}_i)} \mathbb{I}(f(x_i) = y_i) - \hat{p}(x_i) \right|$$

Note that there are two approximations: we first discretize the ECE, and then use finite data to approximate the discretized expression

$$\mathrm{ECE}(f, \hat{p}) \approx \mathrm{ECE}(f, \hat{p}; \mathbf{c}) \approx \mathrm{E\hat{C}E}(f, \hat{p}; \mathbf{c}, \mathcal{D})$$

In practice, if the first approximation is better (more bins are used), then the second approximation must be worse (there will be less data in each bin) (Kumar et al., 2019). In other words, with finite data, there is a tradeoff between calibration error and estimation error. Note that newer estimators, e.g. (Kumar et al., 2019), can measure the ECE even more accurately, particularly when there are more bins.

## A.2 LAPLACE MECHANISM PROOF

**Theorem 1.** *The Laplace mechanism (Dwork et al., 2014) preserves $\epsilon$-differential privacy.*

*Proof.* Let $D \in \mathbb{N}^{|\mathcal{X}|}$ and $D' \in \mathbb{N}^{|\mathcal{X}|}$ be two databases that differ by up to one element, i.e. $\|D - D'\|_1 \leq 1$. Let function $f : \mathbb{N}^{|\mathcal{X}|} \to \mathbb{R}^z$, and let $p_D$ and $p'_D$ denote the probability density functions of $\mathcal{M}_L(D; f, \epsilon)$ and $\mathcal{M}_L(D'; f, \epsilon)$, respectively. Then we can take the ratio of $p_D$ to $p'_D$ at an arbitrary point $x \in \mathbb{R}^z$:

$$\frac{p_D(x)}{p'_D(x)} = \prod_{i=1}^{z} \left( \frac{\exp(\frac{-\epsilon|f(D)_i - x_i|}{\Delta f})}{\exp(\frac{-\epsilon|f(D')_i - x_i|}{\Delta f})} \right)$$

$$= \prod_{i=1}^{z} \exp\left( \frac{\epsilon(|f(D')_i - x_i| - |f(D)_i - x_i|)}{\Delta f} \right)$$

$$\leq \prod_{i=1}^{z} \exp\left( \frac{\epsilon|f(D)_i - f(D')_i|}{\Delta f} \right)$$

$$= \exp\left( \frac{\epsilon \cdot \|f(D) - f(D')\|_1}{\Delta f} \right)$$

$$\leq \exp(\epsilon)$$

where the first inequality follows from the triangle inequality, and the second inequality follows from the definition of sensitivity (Dwork et al., 2014). □

## B   PROOFS

*Proof of Proposition 1.*

$$\frac{\partial}{\partial T}\mathbb{E}_{x,y\sim p^*}\left[\log\frac{e^{l_y(x)/T}}{\sum_j e^{l_j(x)/T}}\right] = \mathbb{E}_{x,y\sim p^*}\left[\frac{\partial}{\partial T}l_y(x)/T - \frac{\partial}{\partial T}\log\sum_j e^{l_j(x)/T}\right]$$

$$= \mathbb{E}_{x,y\sim p^*}\left[-l_y(x)/T^2 - \frac{-\sum_j e^{l_j(x)/T}l_j(x)/T^2}{\sum_j e^{l_j(x)/T}}\right]$$

$$= \frac{1}{T^2}\mathbb{E}_{x,y\sim p^*}\left[-l_y(x) + \frac{\sum_j l_j(x)e^{l_j(x)/T}}{\sum_j e^{l_j(x)/T}}\right]$$

Let us set the derivative equal to 0. Suppose there are multiple solutions $T_1 > T_2$; this implies that

$$\mathbb{E}_{x,y\sim p^*}\left[\frac{\sum_j l_j(x)e^{l_j(x)/T_1}}{\sum_j e^{l_j(x)/T_1}}\right] = \mathbb{E}_{x,y\sim p^*}\left[\frac{\sum_j l_j(x)e^{l_j(x)/T_2}}{\sum_j e^{l_j(x)/T_2}}\right]. \tag{5}$$

$\mathbb{E}_{x,y\sim p^*}\left[\frac{\sum_j l_j(x)e^{l_j(x)/T}}{\sum_j e^{l_j(x)/T}}\right]$ is monotonically non-increasing. Therefore, if there are 0 or 1 solutions to Eq. 5, the original function must be unimodal. If there are at least 2 solutions $T_1 < T_2$, then $\mathbb{E}_{x,y\sim p^*}\left[\frac{\sum_j l_j(x)e^{l_j(x)/T}}{\sum_j e^{l_j(x)/T}}\right]$ must be a constant function $\forall T \in [T_1, T_2]$, which implies that $\mathbb{E}_{x,y\sim p^*}\left[\frac{e^{l_y(x)/T}}{\sum_j e^{l_j(x)/T}}\right]$ is a constant function of $T \in [T_1, T_2]$. This further implies that $\mathbb{E}_{x,y\sim p^*}\left[\frac{e^{l_y(x)/T}}{\sum_j e^{l_j(x)/T}}\right]$ is a constant function for all $T \in \mathbb{R}$, which is also unimodal. □

*Proof of Proposition 2.* Because $\hat{p}_T(x)$ is a monotonically decreasing function of $T$, $\mathbb{E}_{p^*}[\hat{p}_T(x)]$ is also a monotonically decreasing function of $T$. This means that $\Pr_{x,y\sim p^*}[f(x) = y] - \mathbb{E}_{p^*}[\hat{p}_T(x)]$ is a monotonically decreasing function of $T$. The absolute value of a monotonic function must be monotonic or unimodal. □

## C   ADDITIONAL DETAILS FOR SECTION 3

### C.1   GOLDEN SECTION SEARCH

The golden section search is an algorithm for finding the extremum of a unimodal function within a specified interval. It is an iterative method that reduces the search interval with each iteration. The algorithm is described below. Note that we describe the algorithm for a minimization problem, but it also works for maximization problems.

1. Specify the function to be minimized, $g(\cdot)$, and specify an interval over which to minimize $g$, $[T_{min}, T_{max}]$.

2. Select two interior points $T_1$ and $T_2$, with $T_1 < T_2$, such that $T_1 = T_{max} - \frac{\sqrt{5}-1}{2}(T_{max} - T_{min})$ and $T_2 = T_{min} + \frac{\sqrt{5}-1}{2}(T_{max} - T_{min})$. Evaluate $g(T_1)$ and $g(T_2)$.

3. If $g(T_1) > g(T_2)$, then determine a new $T_{min}, T_1, T_2, T_{max}$ as follows:

$$T_{min} = T_1$$
$$T_{max} = T_{max}$$
$$T_1 = T_2$$
$$T_2 = T_{min} + \frac{\sqrt{5}-1}{2}(T_{max} - T_{min})$$

If $g(T_1) < g(T_2)$, determine a new $T_{min}, T_1, T_2, T_{max}$ as follows:

$$T_{min} = T_{min}$$
$$T_{max} = T_2$$
$$T_2 = T_1$$
$$T_1 = T_{max} - \frac{\sqrt{5} - 1}{2}(T_{max} - T_{min})$$

Note that in either case, only one new calculation is performed.

4. If the interval is sufficiently small, i.e. $T_{max} - T_{min} < \delta$, then the maximum occurs at $(T_{min} + T_{max})/2$. Otherwise, repeat Step 3.

### C.2  ADAPTING EXISTING RECALIBRATION METHODS FOR DIFFERENTIAL PRIVACY

In this section, we go into more detail about how to adapt several existing recalibration algorithms for the differential privacy setting with our framework.

#### C.2.1  TEMPERATURE SCALING

Temperature scaling optimizes over the temperature parameter $T$ using the negative log likelihood loss, and thus requires multiple iterations to query the databases $D_i$ at different temperature values using golden section search. In this case, the objective function $g(\cdot)$ is the negative log-likelihood (NLL) loss over all samples. In the standard NLL formulation, the overall loss is the average of the samples' NLL losses, but summing these losses for each database rather than taking the average is equivalent except for a constant scale factor (the total number of samples in the database). Thus, the function $f^k$ queries each $D_i$ for its summed NLL loss. The sensitivity $\Delta f$ is technically infinite, since the range of the NLL function is infinite, but in practice we can choose some sufficiently large value (we chose $\Delta f = 10$, since that was approximately the largest NLL value we saw among the images that we checked). We chose $T_{min} = 0.5$ and $T_{max} = 3.0$, since empirically the optimal temperature always seems to fall within this range, and used $K = 5$ iterations. To aggregate information from different $D_i$, we simply average the $\mathcal{M}^k(D_1), \cdots, \mathcal{M}^k(D_d)$. The new classifier $(\phi, \hat{p}')$ outputs probabilities that are recalibrated with the (noisy) optimal temperature.

#### C.2.2  TEMPERATURE SCALING BY ECE MINIMIZATION

The standard recalibration objective when applying temperature scaling is to maximize the log likelihood of a validation dataset. This objective is given in both recent papers (Guo et al., 2017) and established textbooks (Smola et al., 2000). An alternative, but surprisingly overlooked, objective is to minimize the discretized ECE directly. To adapt this method to differential privacy, we must again use multiple iterations to query the databases $D_i$ at different temperature values using golden section search. Here we want to find the temperature that minimizes the discretized ECE:

$$\min_T \sum_{bins} |\text{Acc} - \text{Conf}| \cdot pr = \min_T \sum_{bins} \left| \frac{n_{correct}}{n_{bin}} - \frac{\sum_i c_i}{n_{bin}} \right| \cdot \frac{n_{bin}}{n_{total}} \tag{6}$$

where $pr$ is the proportion of samples in the bin, $n_{correct}$ is the number of correct predictions in the bin, $n_{bin}$ is the total number of samples in the bin, $\sum_i c_i$ is the sum of the confidence scores for all samples in the bin, and $n_{total}$ is the total number of samples across all bins.

Simplifying Eq. 6 and ignoring $n_{total}$ as a constant, our objective function $g(\cdot)$ becomes

$$g(\phi, T) = \sum_{bins} |n_{correct} - \sum_i c_i|$$

The function $f^k$ queries each $D_i$ for the quantity $(n_{correct} - \sum_i c_i)$ in each bin. The sensitivity $\Delta f = 1$, since this quantity could change by up to 1 with the addition or removal of one sample to a database. We use $T_{min} = 0.5$, $T_{max} = 3.0$, and $K = 5$ iterations. We use 15 bins (since we also evaluate the discretized ECE with 15 bins), so the $\mathcal{M}^k$ are vectors $\in \mathbb{R}^{15}$. To aggregate information from different $D_i$, we average the $\mathcal{M}^k(D_1), \cdots, \mathcal{M}^k(D_d)$, take the absolute value of this average, and then sum this absolute value vector over all bins. In the absence of noise, this aggregation process

will yield the correct overall $g(\cdot)$ exactly, using all samples from all sources. The new classifier $(\phi, \hat{p}')$ outputs probabilities that are recalibrated with the (noisy) optimal temperature. Unsurprisingly, ECE-T performs very well without the constraints of differential privacy, so this method may be a good choice when $\epsilon$ is high.

### C.2.3 HISTOGRAM BINNING

Histogram binning is a relatively simple, non-parametric recalibration method that can be adapted to differential privacy with a single iteration (i.e. $K = 1$). The function $f^1$ queries $D_i$ for the number of correct predictions in each bin and the total number of samples in each bin. $\Delta f = 2$ because if exactly one entry is added or removed from a database, the number of correct predictions can change by at most 1 for exactly one of the bins, and the total number of samples can change by at most 1 for exactly one of the bins. We use 15 bins in our experiments, so the $\mathcal{M}^k$ are vectors $\in \mathbb{R}^{30}$. To aggregate information from different $D_i$, we average the $\mathcal{M}^k(D_1), \cdots, \mathcal{M}^k(D_d)$. The new confidence for each bin is the average number of correct predictions divided by the average total number of samples for that bin.

## D  ADDITIONAL DETAILS FOR SECTION 4

### D.1  FACTORS THAT AFFECT CALIBRATION QUALITY AND PRIVACY

Table 2: The impact of various factors on recalibration quality and privacy preservation.

|  | ↑ Data | ↑ Iterations | ↑ Bins | ↑ Sensitivity | ↑ $\epsilon$ |
|---|---|---|---|---|---|
| Calibration quality | ↗ | ↗ | ↗ | -- | -- |
| Privacy preservation | ↗ | ↘ | ↘ | ↘ | ↘ |

Table 2 shows several factors and hyperparameter choices that affect the calibration quality and the level of privacy for all recalibration methods. More data improves both calibration and privacy. More iterations improves calibration when privacy is not required (e.g. running more iterations of gradient descent), but hurts privacy (making multiple queries in a parametric optimization setting with the same amount of added noise increases $\epsilon$). Using more bins for methods that involve binning improves calibration when enough data is available, but may hurt privacy. Higher sensitivity of the $f^k$ function hurts privacy, and higher $\epsilon$ represents less privacy. We discuss each of these in more detail below.

**Data**  Differentially private recalibration algorithms require sufficient data in order to work well. We cannot trivially combine data from different private datasets because each dataset holder must honor its agreement with the individuals whose information is in that dataset. Our framework describes a method for pooling data from different private datasets while allowing each one to respect differential privacy for its users, which is necessary for improved calibration while preserving privacy.

**Number of iterations**  For parametric optimization recalibration methods, multiple iterations are generally needed to search the parameter space. Using additional iterations improves the calibration without differential privacy (e.g. running more iterations of gradient descent), but hurts the calibration when differential privacy is required. With multiple iterations, a worst-case bound on the overall $L_1$ sensitivity of $f^k$ is $K$ times the sensitivity of a single query $\Delta f_{single}$, since a single database entry may change the response to each query by up to $\Delta f_{single}$. Thus, the amount of noise added to the true query responses must follow a $L(0, K \cdot \Delta f_{single}/\epsilon)$ distribution to maintain $\epsilon$-differential privacy. Because using more iterations increases the amount of noise added, it is best to search through the parameter space while minimizing the number of iterations needed for the desired granularity. We use golden section search to do this. Each iteration of the golden section search narrows the range of possible values of the extremum, but increases the amount of noise added to the data; in general, we select $K$ such that the granularity and the noise are balanced.

**Binning**  Several of the recalibration methods discussed use binning, where all of the confidence estimates are divided into mutually-exclusive bins. Without differential privacy, using more bins

generally improves calibration when a lot of data is available (i.e. above a "data threshold"), but hurts calibration below this data threshold. When not enough data is available, using more bins increases the estimation error since there are too few samples in each bin. In the differential privacy setting, using more bins may degrade the calibration. In this setting, one query may request a summary statistic from each bin. Because a single database entry can be in exactly one bin, the remaining bins are unaffected and the sensitivity does not increase with more bins. However, although the number of bins does not affect the absolute amount of noise, it can affect the relative amount of noise. When more bins are used, there are fewer elements in each bin on average. Thus, the summary statistics involved tend to be lower, and the noise is relatively higher.

Note that when multiple equal-width bins are involved, as in temperature scaling by ECE minimization (see Section C.2.2), the optimization problem may not be strictly unimodal since samples can change bins as the temperature changes. Using bins with equal numbers of samples, rather than equal widths, ensures unimodality in temperature scaling but makes it difficult to combine information from different private data sources (since different sources will have different bin endpoints). Thus, we elected to use equal-width bins in our experiments. Although the function to be minimized is not necessarily unimodal, it is generally a close enough approximation that golden section search returns reasonably good results with few queries, and empirically performs better than grid search.

**Sensitivity of $f^k$**   An $f^k$ function with a large range has a detrimental effect on the amount of noise added. For instance, the range of the negative log-likelihood is technically infinite (although in practice we used some sufficiently large value). Thus, the sensitivity of a method with the negative log-likelihood in the objective function is quite high, and the amount of noise needed to preserve differential privacy is large.

**$\epsilon$ value**   Calibration is worse when $\epsilon$ is smaller, i.e. when there is a higher privacy level with stronger differential privacy constraints.

# E   ADDITIONAL EXPERIMENTAL DETAILS AND RESULTS

## E.1   EXPERIMENTAL SETUP

We simulated the problem of recalibration with multiple private datasets on domain-shifted data using the ImageNet-C, CIFAR-100-C, and CIFAR-10-C datasets (Hendrycks & Dietterich, 2019), which are perturbed versions of the ImageNet (Deng et al., 2009), CIFAR-100 (Krizhevsky & Hinton, 2009), and CIFAR-10 (Krizhevsky & Hinton, 2009) test sets respectively. We randomly divided each perturbed test set into $n_{sources}$ validation sets of size $n_{samples}$ and a test set comprising the remaining images, where $n_{sources}$ represents the number of private data sources and $n_{samples}$ represents the number of samples per source. We computed each ECE value by binning with 15 equal-width bins.

For ImageNet, we varied the number of private data sources from 100 to 2000 in step sizes of 100, with 10 samples per data source and $\epsilon = 1$. We varied the number of samples per data source from 5 to 100 in step sizes of 5, with 100 private data sources and $\epsilon = 1$. We varied $\epsilon$ from 0.2 to 2.0 in step sizes of 0.2, with 50 samples per data source and 100 private data sources. For CIFAR-100 and CIFAR-10, we varied the number of private data sources from 10 to 250 in step sizes of 10, with 10 samples per data source and $\epsilon = 1$. We varied the number of samples per data source from 5 to 50 in step sizes of 5, with 50 private data sources and $\epsilon = 1$. We varied $\epsilon$ from 0.2 to 2.0 in step sizes of 0.2, with 30 samples per data source and 50 private data sources. We used $K = 5$ iterations for all experiments. We reported the average ECE achieved over 500 randomly divided trials for each experiment.

All models were trained on only the unperturbed training sets. For ImageNet, we trained a ResNet50 network (He et al., 2015) for 90 epochs with an SGD optimizer (Sutskever et al., 2013) with an initial learning rate of 0.1, and decayed the learning rate according to a cosine annealing schedule (Loshchilov & Hutter, 2016). For CIFAR-100 and CIFAR-10, we trained Wide ResNet-28-10 networks (Zagoruyko & Komodakis, 2016) for 200 epochs with an SGD optimizer with an initial learning rate of 0.1, and again decayed the learning rate with a cosine annealing schedule. For each dataset, we tested both the unperturbed accuracy and the perturbed accuracy on each of 15

perturbation types in (Hendrycks & Dietterich, 2019) at multiple severity levels to ensure sharpness. These accuracy tables can be found in E.2.

## E.2 EXPERIMENTS WITHOUT DIFFERENTIAL PRIVACY CONSTRAINTS

| | **Classification Accuracy** | | |
|---|---|---|---|
| Perturbation Type | CIFAR-10 | CIFAR-100 | ImageNet |
| Brightness | 0.9290 | 0.7107 | 0.5570 |
| Contrast | 0.4656 | 0.2967 | 0.0422 |
| Defocus Blur | 0.6402 | 0.4008 | 0.1506 |
| Elastic Transform | 0.7616 | 0.5214 | 0.1477 |
| Fog | 0.7639 | 0.4808 | 0.2270 |
| Frost | 0.6907 | 0.4196 | 0.2064 |
| Gaussian Noise | 0.2889 | 0.1046 | 0.0447 |
| Glass Blur | 0.5313 | 0.2212 | 0.0834 |
| Impulse Noise | 0.2940 | 0.0642 | 0.0463 |
| Jpeg Compression | 0.7056 | 0.4190 | 0.3318 |
| Motion Blur | 0.7062 | 0.4997 | 0.1337 |
| Pixelate | 0.5137 | 0.2994 | 0.2260 |
| Shot Noise | 0.3581 | 0.1190 | 0.0507 |
| Snow | 0.7975 | 0.5268 | 0.1594 |
| Zoom Blur | 0.7163 | 0.4708 | 0.2287 |
| Unperturbed | 0.9613 | 0.8050 | 0.7613 |

Table 3: Classification accuracies for CIFAR-10, CIFAR-100, and ImageNet under the highest severity perturbations of the CIFAR-10-C, CIFAR-100-C, and ImageNet-C test sets. The classification models used achieve the expected state-of-the-art results for accuracy on the unperturbed test sets.

| **CIFAR-10** | | | | |
|---|---|---|---|---|
| Perturbation Severity = 5 | Base | NLL-T | Acc-T | ECE-T |
| Brightness | 0.0456 | 0.0194 | 0.0278 | **0.0182** |
| Contrast | 0.4202 | 0.0503 | **0.0348** | 0.0368 |
| Defocus Blur | 0.2431 | 0.0381 | 0.0372 | **0.0366** |
| Elastic Transform | 0.1555 | 0.0287 | **0.0264** | 0.0319 |
| Fog | 0.1813 | 0.0463 | **0.0433** | 0.0435 |
| Frost | 0.2207 | 0.0636 | **0.0570** | 0.0581 |
| Gaussian Noise | 0.6052 | 0.0624 | **0.0364** | 0.0530 |
| Glass Blur | 0.3434 | 0.0426 | 0.0393 | **0.0389** |
| Impulse Noise | 0.4963 | 0.0570 | **0.0499** | 0.0566 |
| Jpeg Compression | 0.2000 | 0.0423 | 0.0344 | **0.0338** |
| Motion Blur | 0.2153 | 0.0430 | **0.0395** | 0.0427 |
| Pixelate | 0.3840 | 0.0676 | **0.0620** | 0.0649 |
| Shot Noise | 0.5282 | 0.0503 | **0.0464** | 0.0484 |
| Snow | 0.1412 | 0.0390 | **0.0321** | 0.0391 |
| Zoom Blur | 0.1931 | 0.0382 | **0.0343** | 0.0363 |
| Unperturbed | 0.0251 | 0.0078 | 0.0089 | **0.0075** |

Table 4: Expected calibration error (ECE) on CIFAR-10 without privacy constraints is shown. Columns from left to right show ECE for the baseline without calibration, recalibration with temperature scaling by minimizing the negative log likelihood, recalibration with temperature scaling by matching predictive confidence to accuracy (our method), and recalibration by minimizing the ECE directly. Rows indicate the type of perturbation applied. These results correspond to a perturbation severity of 5. Best calibration for each perturbation is shown in **bold**.

**CIFAR-100**

| Perturbation Severity = 5 | Base | NLL-T | Acc-T | ECE-T |
|---|---|---|---|---|
| Brightness | 0.1087 | 0.0602 | 0.0460 | **0.0449** |
| Contrast | 0.3817 | 0.0839 | 0.0574 | **0.0518** |
| Defocus Blur | 0.2707 | 0.0847 | 0.0785 | **0.0780** |
| Elastic Transform | 0.1776 | 0.0626 | **0.0589** | 0.0639 |
| Fog | 0.2217 | 0.0656 | **0.0560** | 0.0574 |
| Frost | 0.2929 | 0.0872 | 0.0791 | **0.0771** |
| Gaussian Noise | 0.6313 | 0.0518 | 0.0423 | **0.0316** |
| Glass Blur | 0.4438 | 0.0777 | **0.0658** | 0.0700 |
| Impulse Noise | 0.3574 | 0.0160 | **0.0061** | 0.0081 |
| Jpeg Compression | 0.2042 | 0.0667 | 0.0512 | **0.0492** |
| Motion Blur | 0.2040 | 0.0707 | 0.0548 | **0.0546** |
| Pixelate | 0.3639 | 0.0599 | **0.0322** | 0.0327 |
| Shot Noise | 0.6200 | 0.0654 | 0.0468 | **0.0391** |
| Snow | 0.1912 | 0.0674 | **0.0575** | 0.0585 |
| Zoom Blur | 0.2189 | 0.0805 | 0.0726 | **0.0724** |
| Unperturbed | 0.0793 | 0.0456 | 0.0319 | **0.0305** |

Table 5: Expected calibration error (ECE) on CIFAR-100 without privacy constraints is shown. Columns from left to right show ECE for the baseline without calibration, recalibration with temperature scaling by minimizing the negative log likelihood, recalibration with temperature scaling by matching predictive confidence to accuracy (our method), and recalibration by minimizing the ECE directly. Rows indicate the type of perturbation applied. These results correspond to a perturbation severity of 5. Best calibration for each perturbation is shown in **bold**.

**ImageNet**

| Perturbation Severity = 5 | Base | NLL-T | Acc-T | ECE-T |
|---|---|---|---|---|
| Brightness | 0.0413 | 0.0325 | **0.0298** | 0.0307 |
| Contrast | 0.0651 | **0.0083** | 0.0083 | 0.0116 |
| Defocus Blur | 0.0618 | 0.0235 | **0.0230** | 0.0239 |
| Elastic Transform | 0.2426 | **0.0287** | 0.0308 | 0.0308 |
| Fog | 0.1572 | 0.0255 | **0.0231** | 0.0232 |
| Frost | 0.1430 | 0.0254 | 0.0253 | **0.0247** |
| Gaussian Noise | 0.1501 | **0.0070** | 0.0080 | 0.0092 |
| Glass Blur | 0.1340 | **0.0160** | 0.0164 | 0.0168 |
| Impulse Noise | 0.1555 | 0.0084 | 0.0069 | **0.0066** |
| Jpeg Compression | 0.0855 | **0.0188** | 0.0228 | 0.0189 |
| Motion Blur | 0.1254 | **0.0180** | 0.0183 | 0.0194 |
| Pixelate | 0.1306 | 0.0175 | 0.0172 | **0.0170** |
| Shot Noise | 0.1820 | 0.0085 | **0.0081** | 0.0109 |
| Snow | 0.1895 | 0.0327 | 0.0323 | **0.0321** |
| Zoom Blur | 0.1343 | 0.0200 | **0.0191** | 0.0193 |
| Unperturbed | 0.0390 | 0.0240 | **0.0239** | 0.0261 |

Table 6: Expected calibration error (ECE) on ImageNet without privacy constraints is shown. Columns from left to right show ECE for the baseline without calibration, recalibration with temperature scaling by minimizing the negative log likelihood, recalibration with temperature scaling by matching predictive confidence to accuracy (our method), and recalibration by minimizing the ECE directly. Rows indicate the type of perturbation applied. These results correspond to a perturbation severity of 5. Best calibration for each perturbation is shown in **bold**.

Table 3 shows the classification accuracy achieved by our models on each of the 15 perturbations of the CIFAR-10-C, CIFAR-100-C, and ImageNet-C test sets, as well as on the unperturbed test set. Note that the models are trained only on unperturbed training data. The accuracies achieved are in line with reported state-of-the-art numbers.

Tables 4, 5, and 6 summarize our calibration results without differential privacy constraints for CIFAR-10, CIFAR-100, and ImageNet, respectively. Our Acc-T algorithm generally improves the model's calibration compared to the standard temperature scaling method NLL-T. Despite its simplicity, Acc-T also performs on par with ECE-T, generally achieving similar ECEs, even when privacy is not required.

### E.3 EXPERIMENTS WITH DIFFERENTIAL PRIVACY CONSTRAINTS

The figures in this section show recalibration results for ImageNet, CIFAR-100, and CIFAR-10 under the highest severity perturbations. In the left panel of each figure, we vary the number of private data sources. In the middle panel, we vary the number of samples per data source. In the right panel, we vary the privacy level $\epsilon$. Our method, Acc-T, generally does best in these settings.

IMAGENET RESULTS

ImageNet, unperturbed

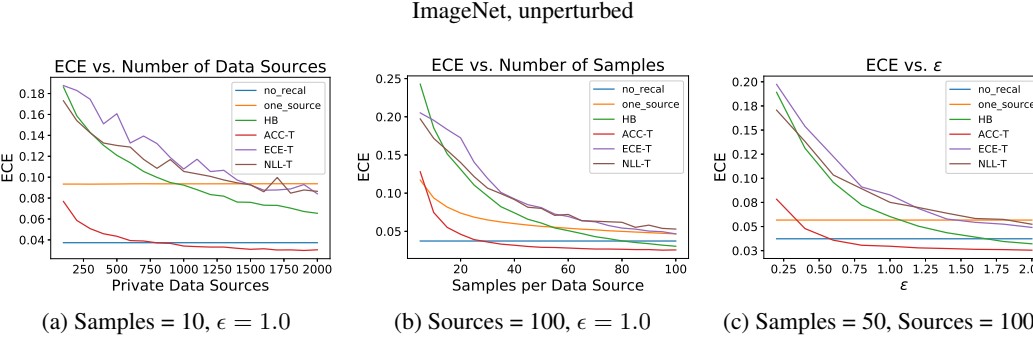

(a) Samples = 10, $\epsilon = 1.0$     (b) Sources = 100, $\epsilon = 1.0$     (c) Samples = 50, Sources = 100

ImageNet, brightness perturbation

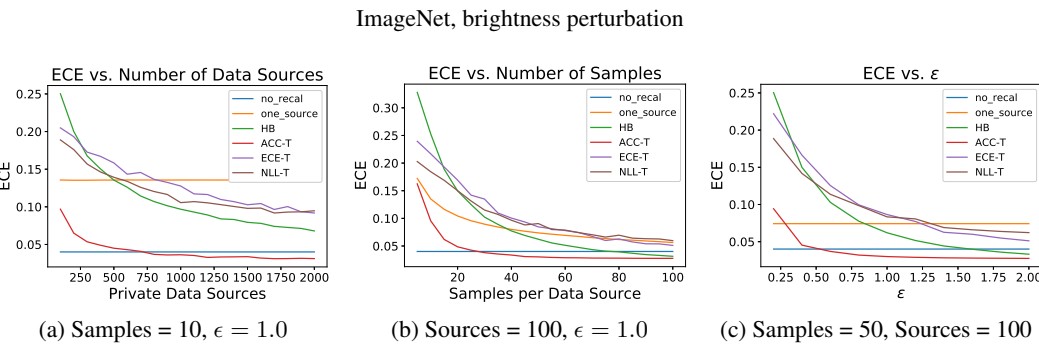

(a) Samples = 10, $\epsilon = 1.0$     (b) Sources = 100, $\epsilon = 1.0$     (c) Samples = 50, Sources = 100

ImageNet, contrast perturbation

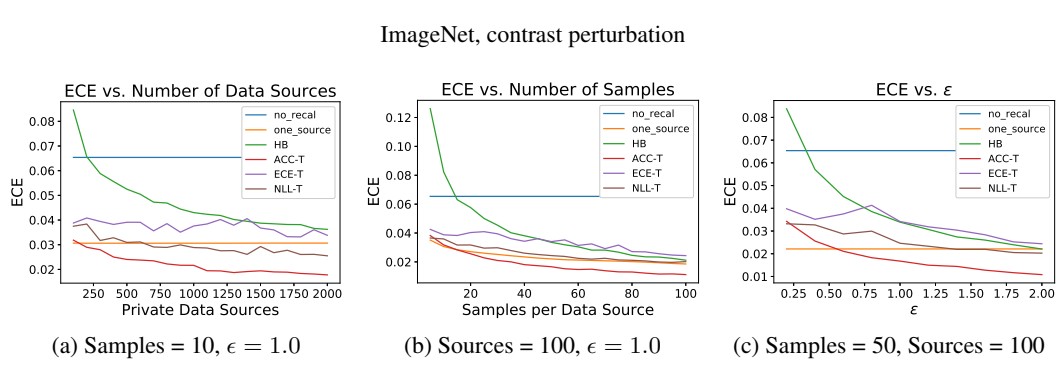

(a) Samples = 10, $\epsilon = 1.0$     (b) Sources = 100, $\epsilon = 1.0$     (c) Samples = 50, Sources = 100

ImageNet, defocus blur perturbation

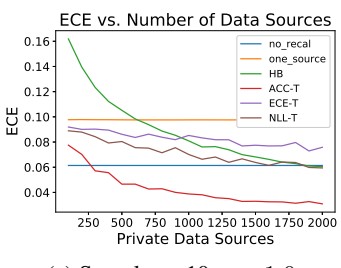
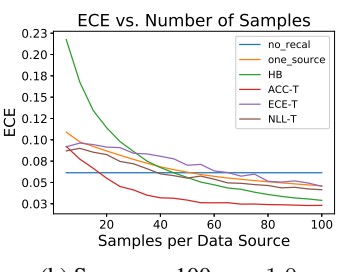
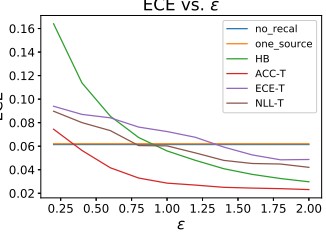

(a) Samples = 10, $\epsilon = 1.0$     (b) Sources = 100, $\epsilon = 1.0$     (c) Samples = 50, Sources = 100

ImageNet, elastic transform perturbation

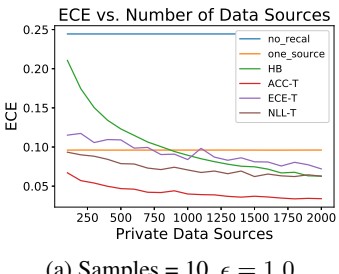
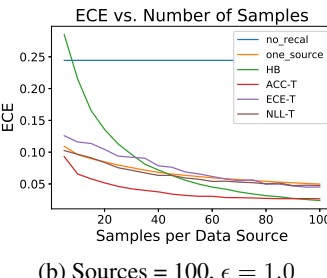
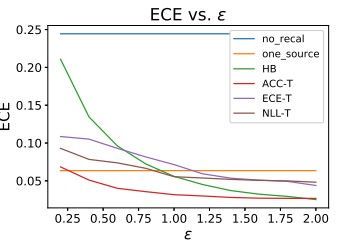

(a) Samples = 10, $\epsilon = 1.0$     (b) Sources = 100, $\epsilon = 1.0$     (c) Samples = 50, Sources = 100

ImageNet, fog perturbation

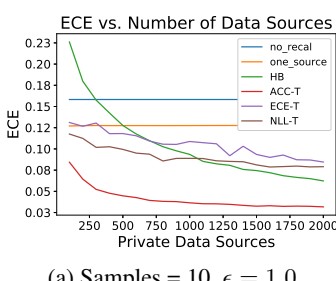
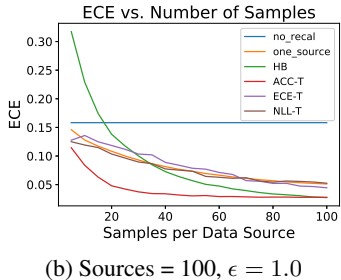
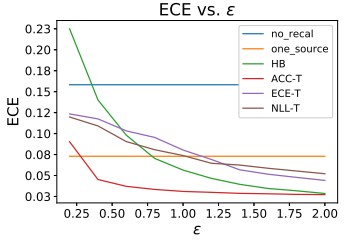

(a) Samples = 10, $\epsilon = 1.0$     (b) Sources = 100, $\epsilon = 1.0$     (c) Samples = 50, Sources = 100

ImageNet, frost perturbation

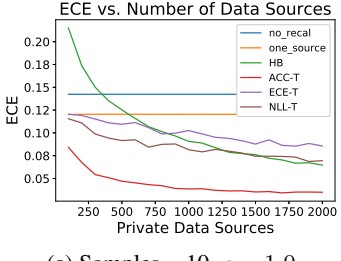
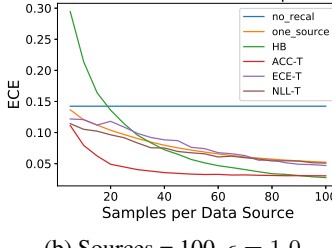
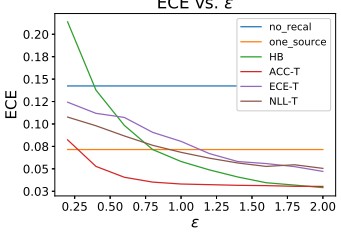

(a) Samples = 10, $\epsilon = 1.0$     (b) Sources = 100, $\epsilon = 1.0$     (c) Samples = 50, Sources = 100

ImageNet, Gaussian noise perturbation

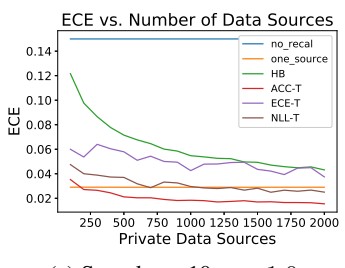

(a) Samples = 10, $\epsilon = 1.0$

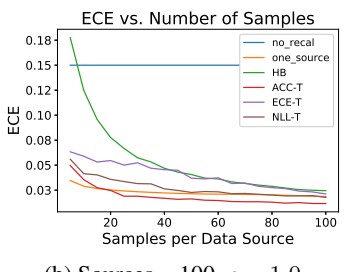

(b) Sources = 100, $\epsilon = 1.0$

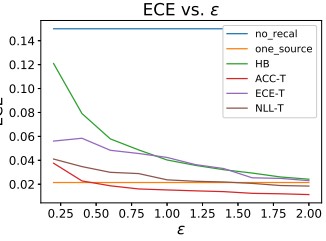

(c) Samples = 50, Sources = 100

ImageNet, glass blur perturbation

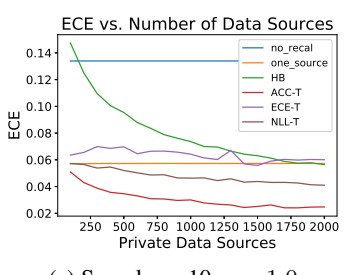

(a) Samples = 10, $\epsilon = 1.0$

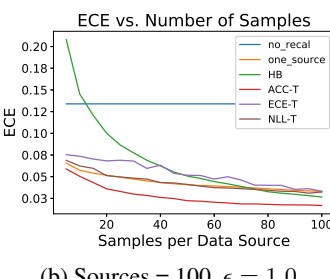

(b) Sources = 100, $\epsilon = 1.0$

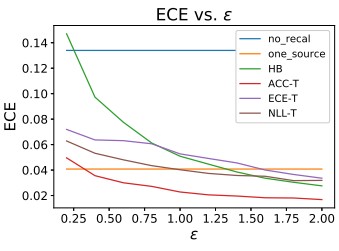

(c) Samples = 50, Sources = 100

ImageNet, impulse noise perturbation

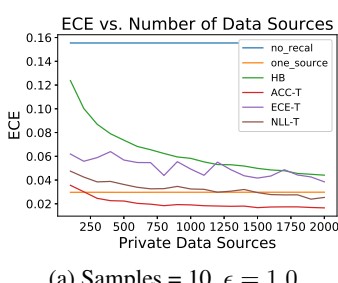

(a) Samples = 10, $\epsilon = 1.0$

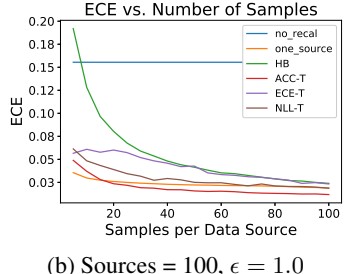

(b) Sources = 100, $\epsilon = 1.0$

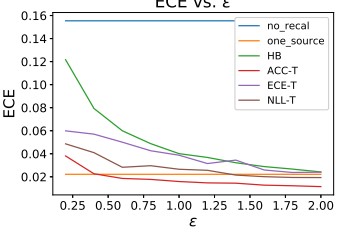

(c) Samples = 50, Sources = 100

ImageNet, jpeg compression perturbation

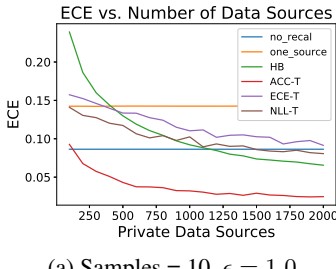

(a) Samples = 10, $\epsilon = 1.0$

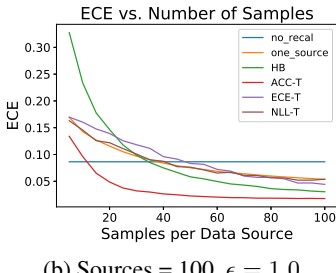

(b) Sources = 100, $\epsilon = 1.0$

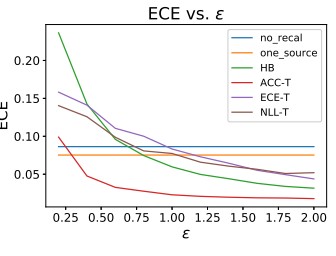

(c) Samples = 50, Sources = 100

ImageNet, motion blur perturbation

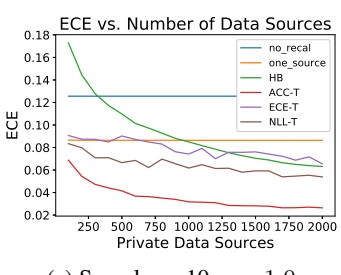

(a) Samples = 10, $\epsilon = 1.0$

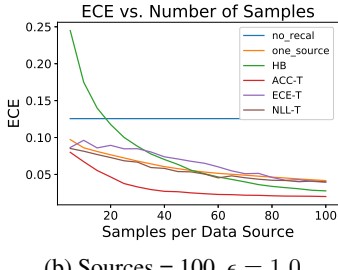

(b) Sources = 100, $\epsilon = 1.0$

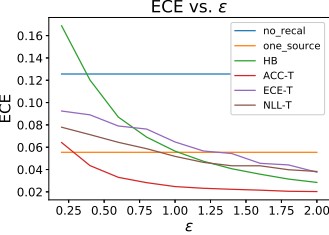

(c) Samples = 50, Sources = 100

ImageNet, pixelate perturbation

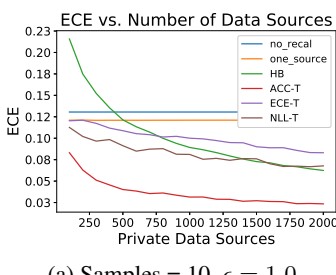

(a) Samples = 10, $\epsilon = 1.0$

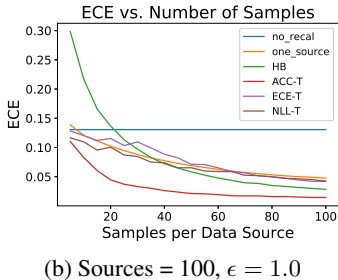

(b) Sources = 100, $\epsilon = 1.0$

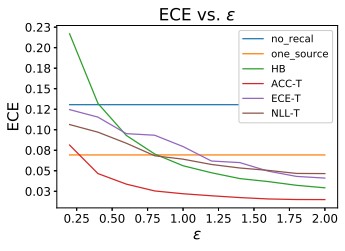

(c) Samples = 50, Sources = 100

ImageNet, shot noise perturbation

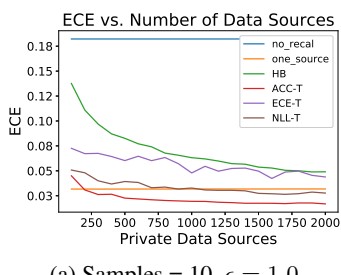

(a) Samples = 10, $\epsilon = 1.0$

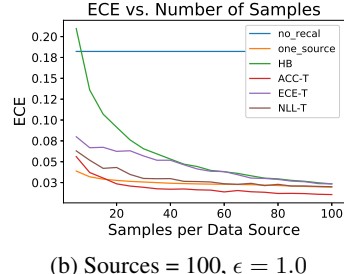

(b) Sources = 100, $\epsilon = 1.0$

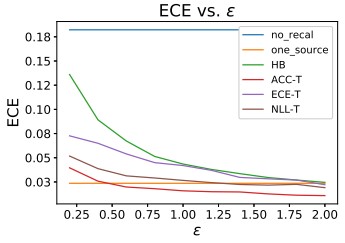

(c) Samples = 50, Sources = 100

ImageNet, snow perturbation

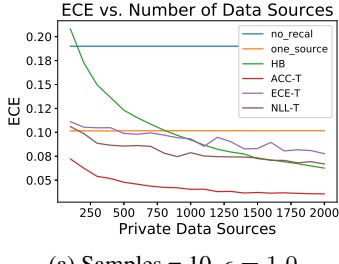

(a) Samples = 10, $\epsilon = 1.0$

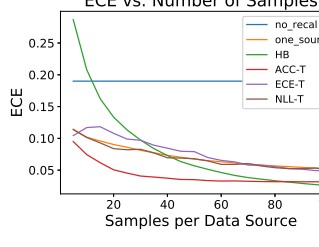

(b) Sources = 100, $\epsilon = 1.0$

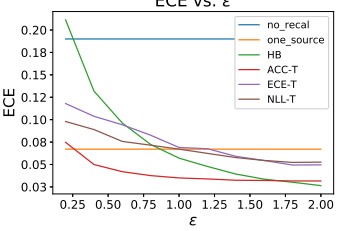

(c) Samples = 50, Sources = 100

ImageNet, zoom blur perturbation

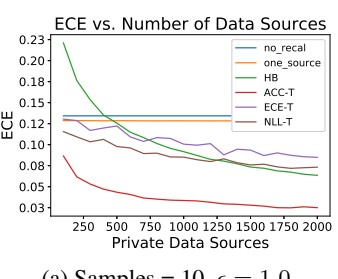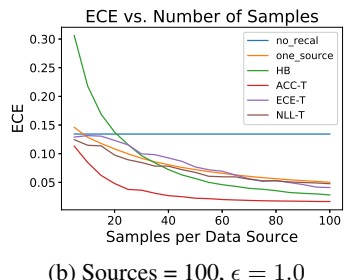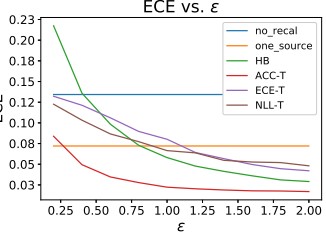

(a) Samples = 10, $\epsilon = 1.0$          (b) Sources = 100, $\epsilon = 1.0$          (c) Samples = 50, Sources = 100

CIFAR-100 RESULTS

CIFAR-100, unperturbed

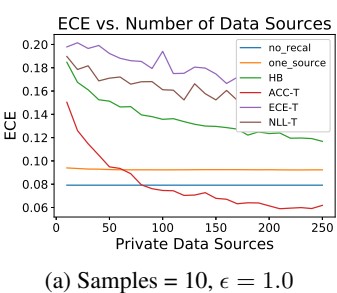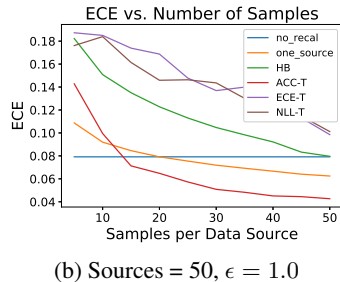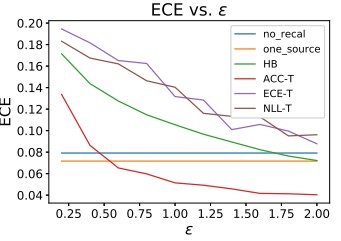

(a) Samples = 10, $\epsilon = 1.0$          (b) Sources = 50, $\epsilon = 1.0$          (c) Samples = 30, Sources = 50

CIFAR-100, brightness perturbation

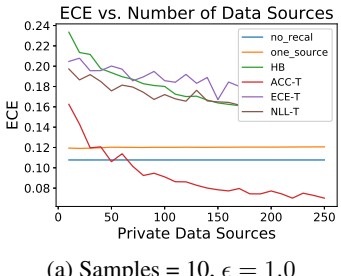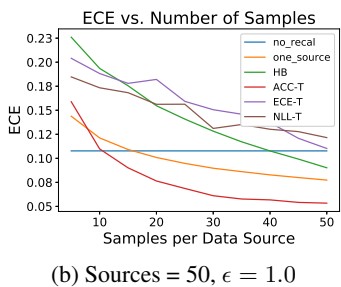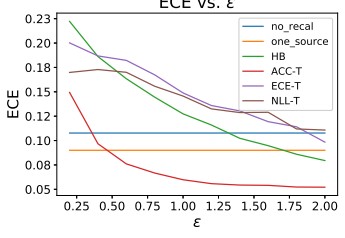

(a) Samples = 10, $\epsilon = 1.0$          (b) Sources = 50, $\epsilon = 1.0$          (c) Samples = 30, Sources = 50

CIFAR-100, contrast perturbation

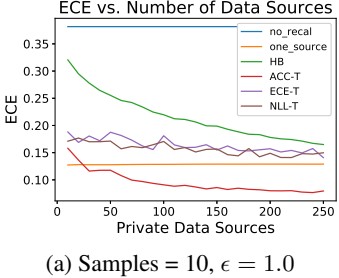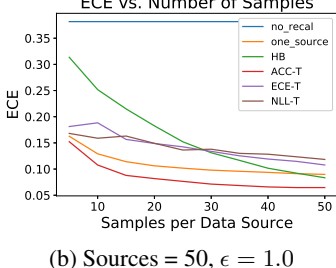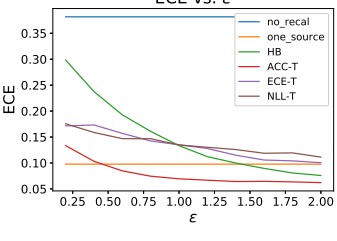

(a) Samples = 10, $\epsilon = 1.0$          (b) Sources = 50, $\epsilon = 1.0$          (c) Samples = 30, Sources = 50

CIFAR-100, defocus blur perturbation

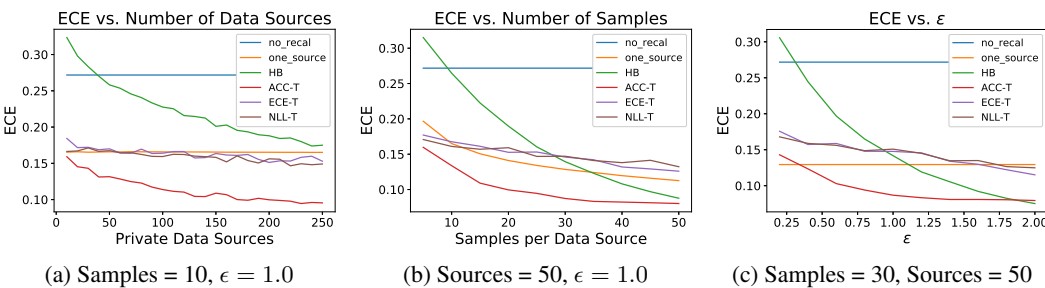

(a) Samples = 10, $\epsilon = 1.0$  (b) Sources = 50, $\epsilon = 1.0$  (c) Samples = 30, Sources = 50

CIFAR-100, elastic transform perturbation

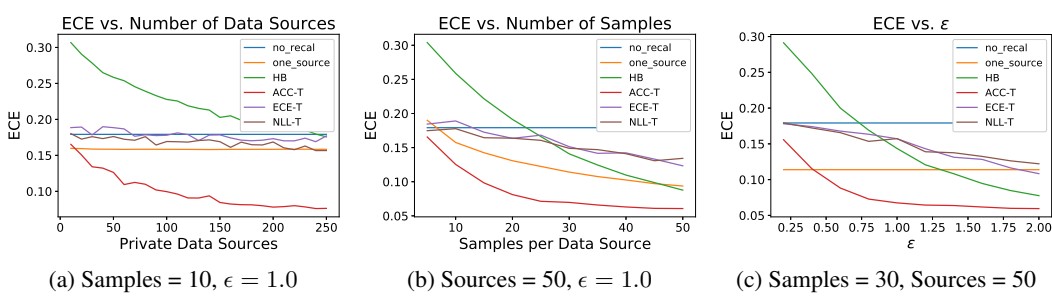

(a) Samples = 10, $\epsilon = 1.0$  (b) Sources = 50, $\epsilon = 1.0$  (c) Samples = 30, Sources = 50

CIFAR-100, fog perturbation

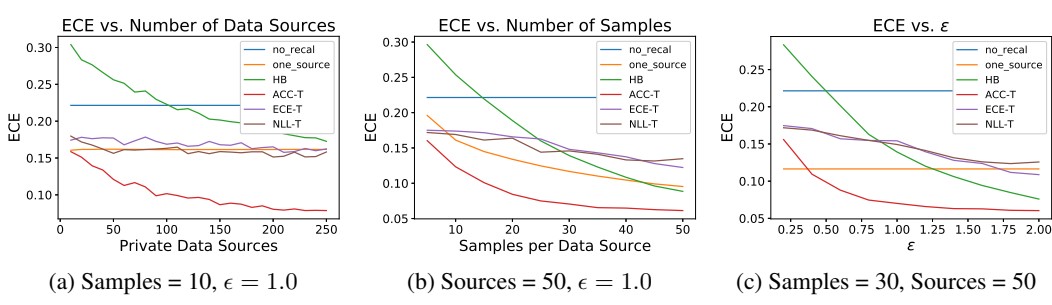

(a) Samples = 10, $\epsilon = 1.0$  (b) Sources = 50, $\epsilon = 1.0$  (c) Samples = 30, Sources = 50

CIFAR-100, frost perturbation

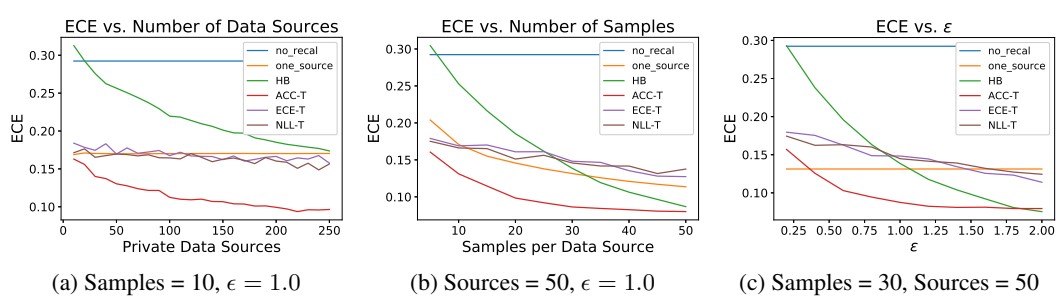

(a) Samples = 10, $\epsilon = 1.0$  (b) Sources = 50, $\epsilon = 1.0$  (c) Samples = 30, Sources = 50

CIFAR-100, Gaussian noise perturbation

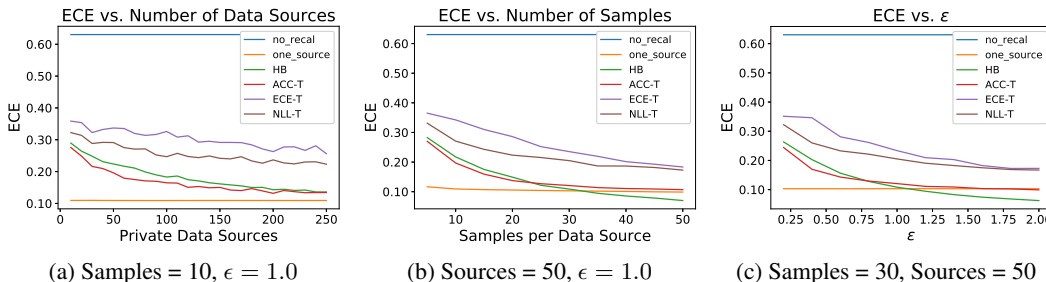

(a) Samples = 10, $\epsilon = 1.0$      (b) Sources = 50, $\epsilon = 1.0$      (c) Samples = 30, Sources = 50

CIFAR-100, glass blur perturbation

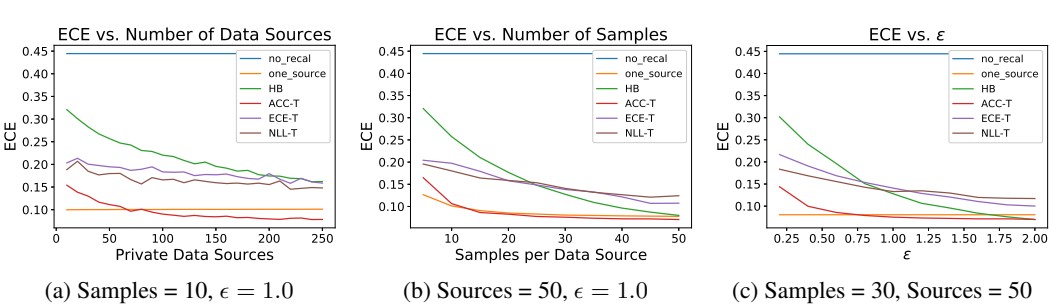

(a) Samples = 10, $\epsilon = 1.0$      (b) Sources = 50, $\epsilon = 1.0$      (c) Samples = 30, Sources = 50

CIFAR-100, impulse noise perturbation

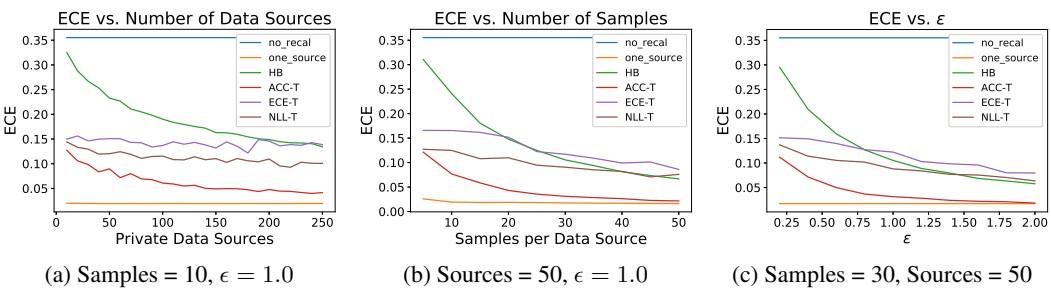

(a) Samples = 10, $\epsilon = 1.0$      (b) Sources = 50, $\epsilon = 1.0$      (c) Samples = 30, Sources = 50

CIFAR-100, jpeg compression perturbation

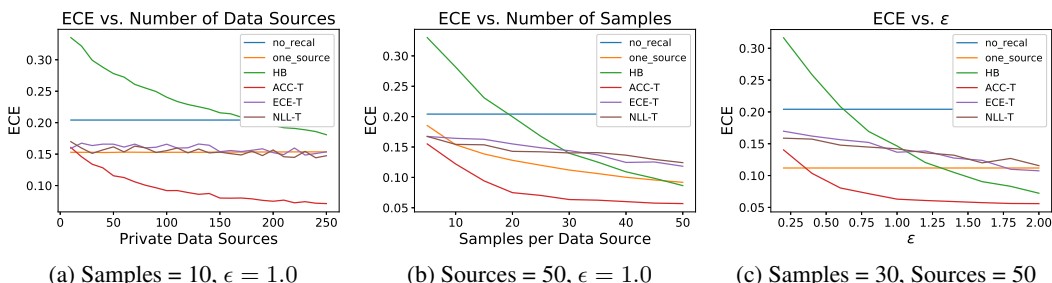

(a) Samples = 10, $\epsilon = 1.0$      (b) Sources = 50, $\epsilon = 1.0$      (c) Samples = 30, Sources = 50

CIFAR-100, motion blur perturbation

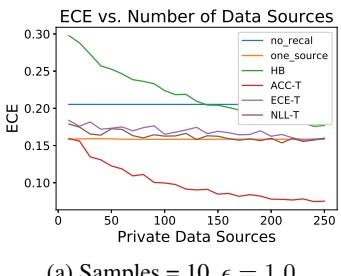 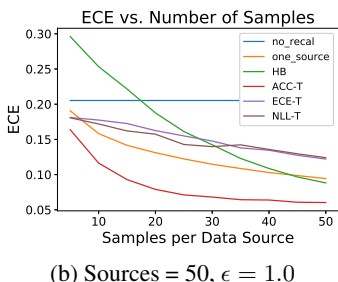 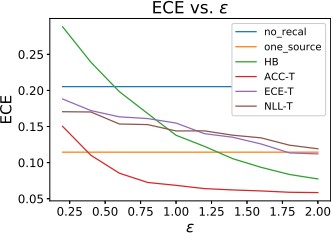

(a) Samples = 10, $\epsilon = 1.0$     (b) Sources = 50, $\epsilon = 1.0$     (c) Samples = 30, Sources = 50

CIFAR-100, pixelate perturbation

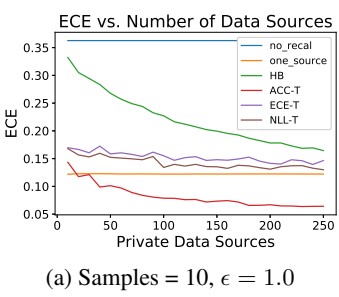 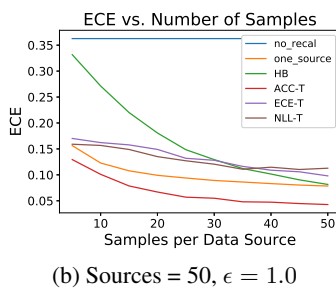 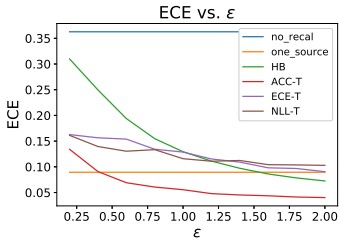

(a) Samples = 10, $\epsilon = 1.0$     (b) Sources = 50, $\epsilon = 1.0$     (c) Samples = 30, Sources = 50

CIFAR-100, shot noise perturbation

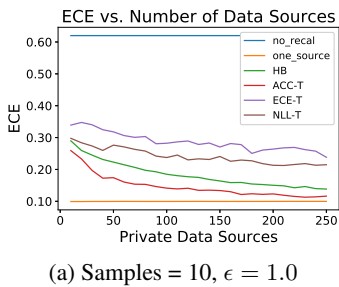 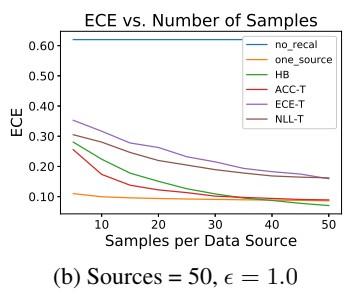 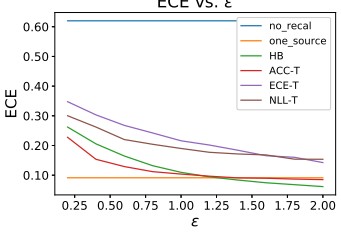

(a) Samples = 10, $\epsilon = 1.0$     (b) Sources = 50, $\epsilon = 1.0$     (c) Samples = 30, Sources = 50

CIFAR-100, snow perturbation

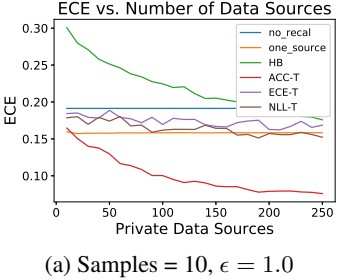 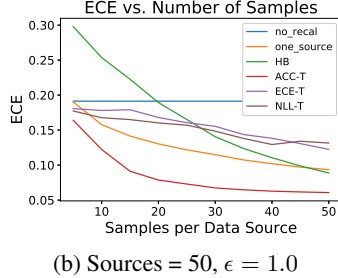 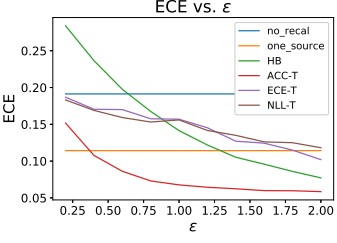

(a) Samples = 10, $\epsilon = 1.0$     (b) Sources = 50, $\epsilon = 1.0$     (c) Samples = 30, Sources = 50

CIFAR-100, zoom blur perturbation

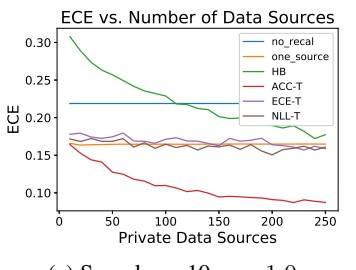 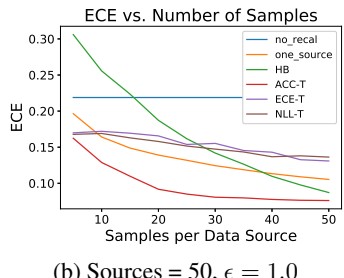 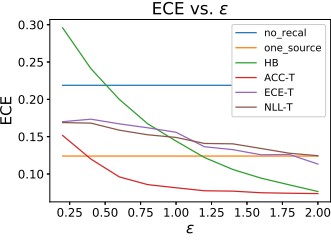

(a) Samples = 10, $\epsilon = 1.0$     (b) Sources = 50, $\epsilon = 1.0$     (c) Samples = 30, Sources = 50

## CIFAR-10 RESULTS

CIFAR-10, unperturbed

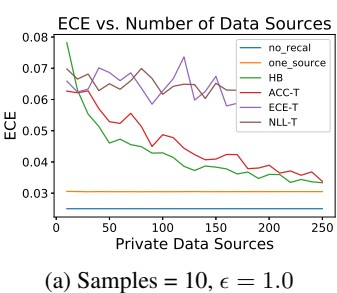 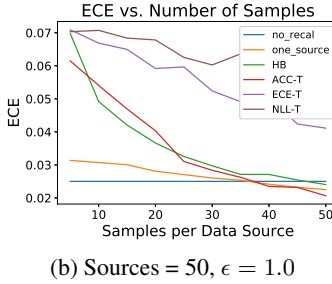 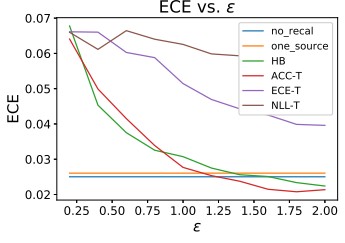

(a) Samples = 10, $\epsilon = 1.0$     (b) Sources = 50, $\epsilon = 1.0$     (c) Samples = 30, Sources = 50

CIFAR-10, brightness perturbation

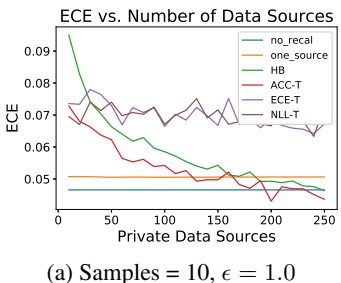 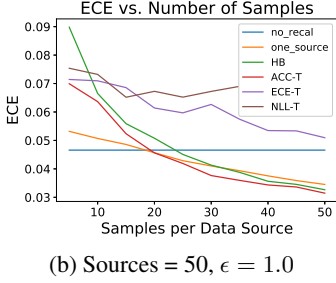 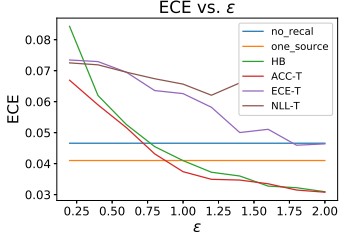

(a) Samples = 10, $\epsilon = 1.0$     (b) Sources = 50, $\epsilon = 1.0$     (c) Samples = 30, Sources = 50

CIFAR-10, contrast perturbation

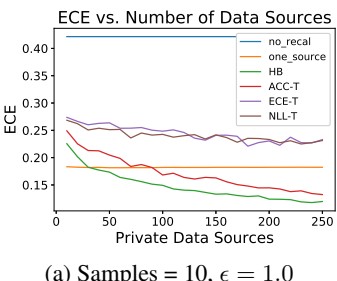 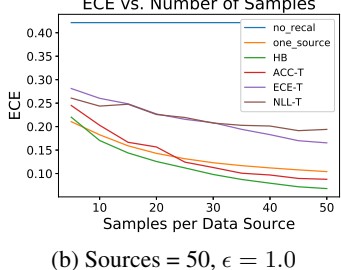 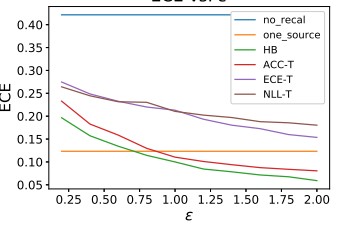

(a) Samples = 10, $\epsilon = 1.0$     (b) Sources = 50, $\epsilon = 1.0$     (c) Samples = 30, Sources = 50

CIFAR-10, defocus blur perturbation

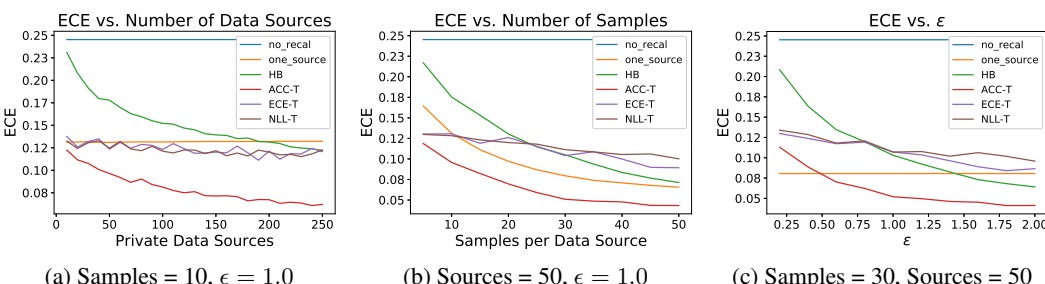

(a) Samples = 10, $\epsilon = 1.0$      (b) Sources = 50, $\epsilon = 1.0$      (c) Samples = 30, Sources = 50

CIFAR-10, elastic transform perturbation

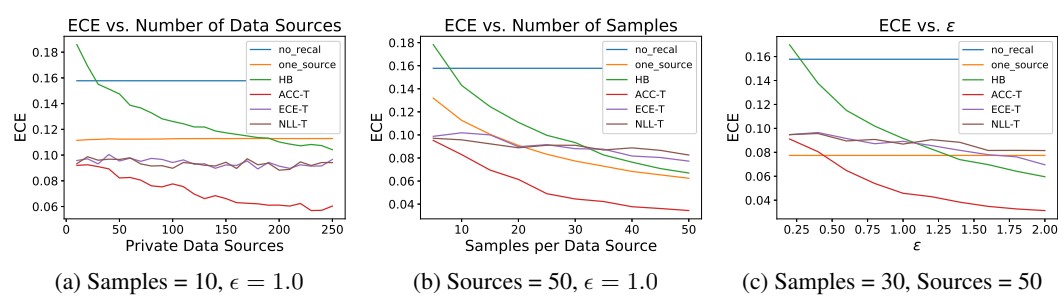

(a) Samples = 10, $\epsilon = 1.0$      (b) Sources = 50, $\epsilon = 1.0$      (c) Samples = 30, Sources = 50

CIFAR-10, fog perturbation

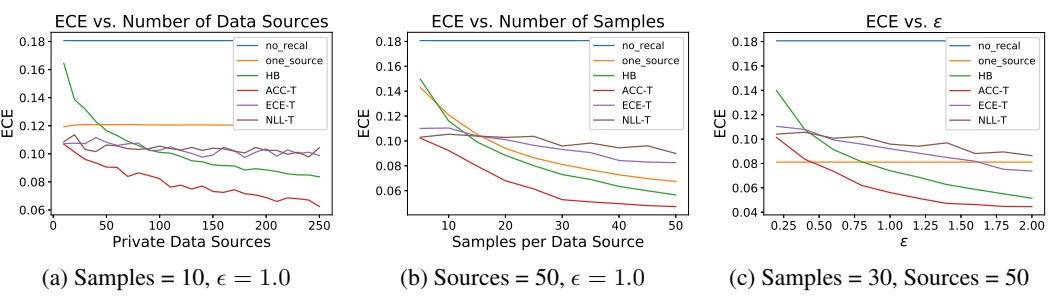

(a) Samples = 10, $\epsilon = 1.0$      (b) Sources = 50, $\epsilon = 1.0$      (c) Samples = 30, Sources = 50

CIFAR-10, frost perturbation

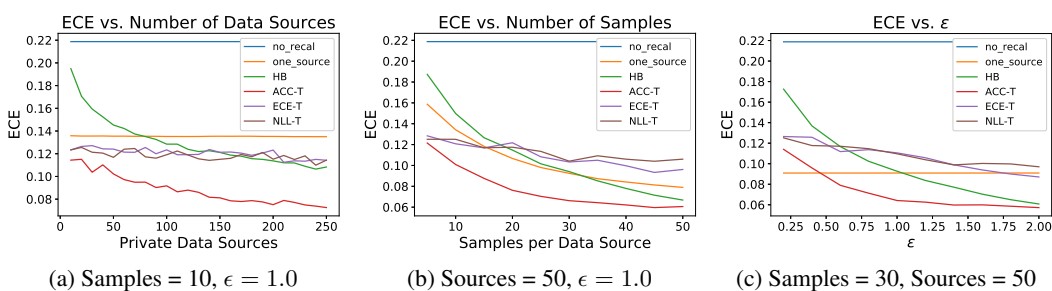

(a) Samples = 10, $\epsilon = 1.0$      (b) Sources = 50, $\epsilon = 1.0$      (c) Samples = 30, Sources = 50

CIFAR-10, Gaussian noise perturbation

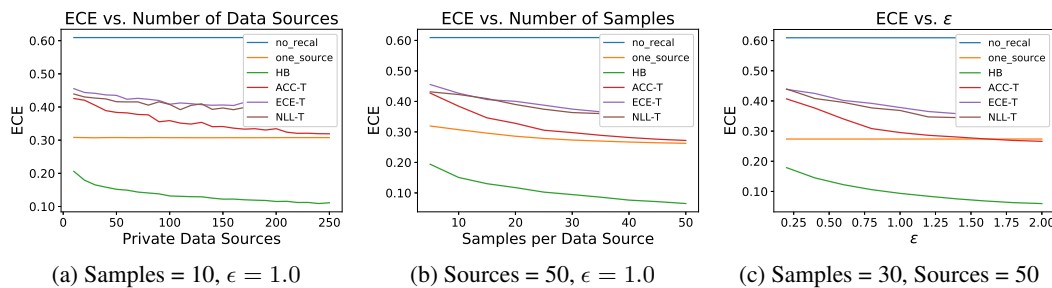

(a) Samples = 10, $\epsilon = 1.0$     (b) Sources = 50, $\epsilon = 1.0$     (c) Samples = 30, Sources = 50

CIFAR-10, glass blur perturbation

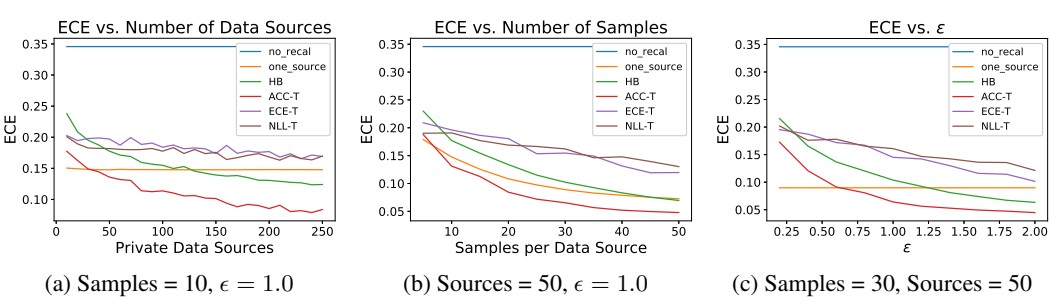

(a) Samples = 10, $\epsilon = 1.0$     (b) Sources = 50, $\epsilon = 1.0$     (c) Samples = 30, Sources = 50

CIFAR-10, impulse noise perturbation

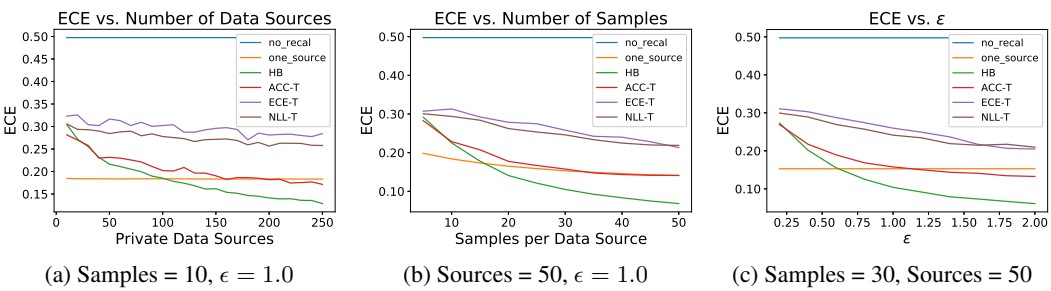

(a) Samples = 10, $\epsilon = 1.0$     (b) Sources = 50, $\epsilon = 1.0$     (c) Samples = 30, Sources = 50

CIFAR-10, jpeg compression perturbation

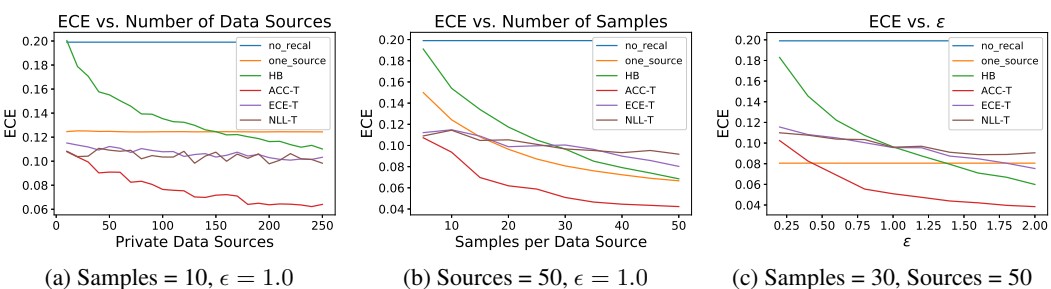

(a) Samples = 10, $\epsilon = 1.0$     (b) Sources = 50, $\epsilon = 1.0$     (c) Samples = 30, Sources = 50

CIFAR-10, motion blur perturbation

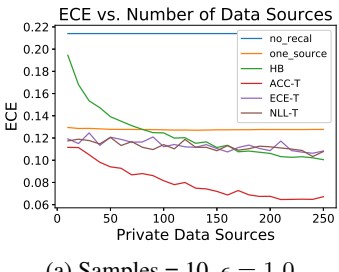 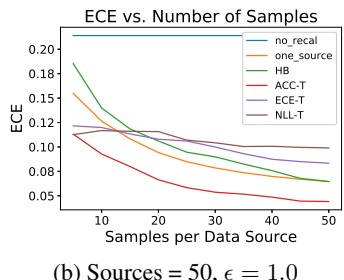 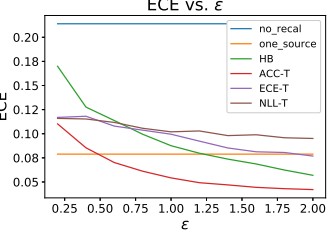

(a) Samples = 10, $\epsilon = 1.0$      (b) Sources = 50, $\epsilon = 1.0$      (c) Samples = 30, Sources = 50

CIFAR-10, pixelate perturbation

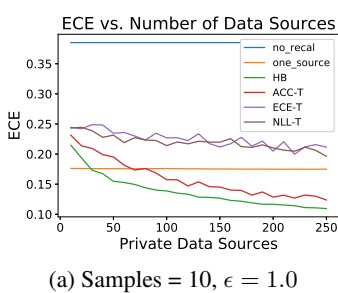 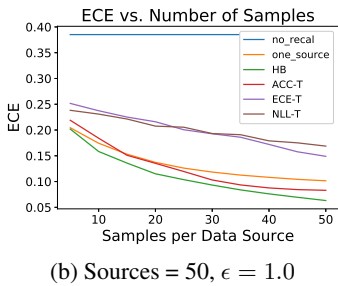 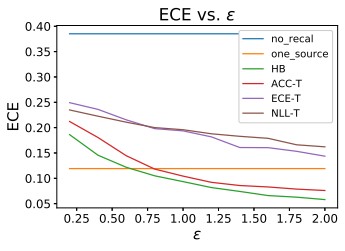

(a) Samples = 10, $\epsilon = 1.0$      (b) Sources = 50, $\epsilon = 1.0$      (c) Samples = 30, Sources = 50

CIFAR-10, shot noise perturbation

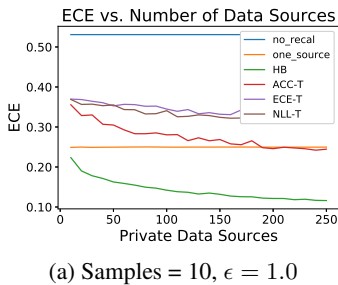 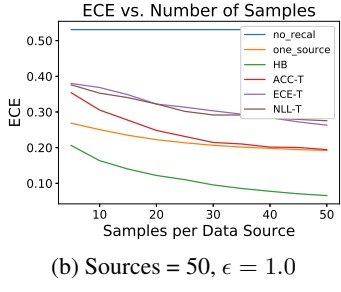 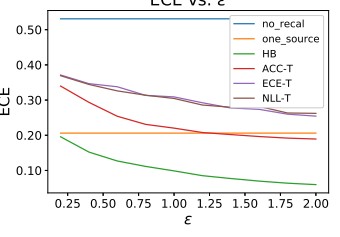

(a) Samples = 10, $\epsilon = 1.0$      (b) Sources = 50, $\epsilon = 1.0$      (c) Samples = 30, Sources = 50

CIFAR-10, snow perturbation

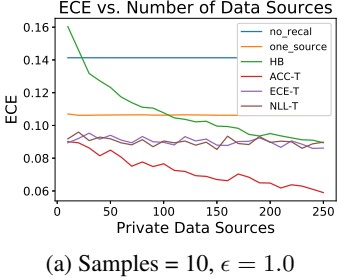 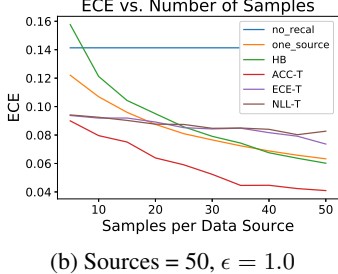 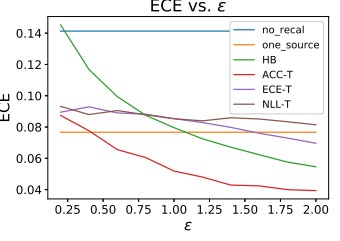

(a) Samples = 10, $\epsilon = 1.0$      (b) Sources = 50, $\epsilon = 1.0$      (c) Samples = 30, Sources = 50

CIFAR-10, zoom blur perturbation

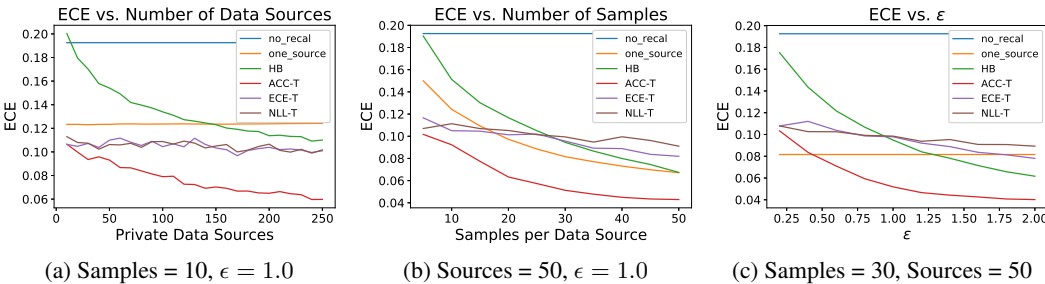

(a) Samples = 10, $\epsilon = 1.0$     (b) Sources = 50, $\epsilon = 1.0$     (c) Samples = 30, Sources = 50

We note that using different clipping thresholds for NLL-T (where the clipped NLL loss is min(clipping_threshold, NLL)) can affect its performance slightly. In practice, selecting the optimal clipping threshold would violate differential privacy, because doing so would require access to the labeled test data. However, even under the most favorable threshold, Acc-T significantly outperforms NLL-T. In Fig. 52, we show an example of NLL-T performance at different clipping thresholds for CIFAR-10 under the "snow" perturbation with a perturbation severity of 1. In this case, using the optimal clipping threshold would improve performance by 0.7% over using a clipping threshold of 10, and this improvement comes at a cost of privacy violations.

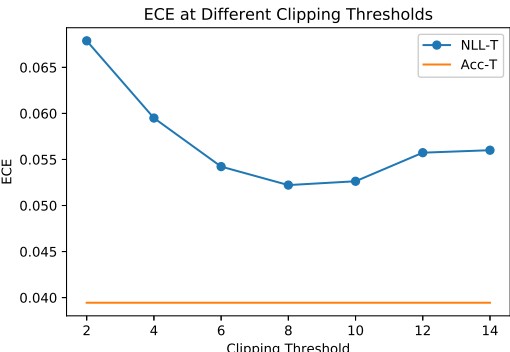

Figure 52: Recalibration results for CIFAR-10 under the "snow" perturbation with a perturbation severity of 1, with different clipping thresholds. Number of sources = 50, number of samples per source = 30, and $\epsilon = 1.0$. Acc-T significantly outperfoms NLL-T regardless of the clipping threshold.

Finally, Table 7 shows the overall median and mean ECE achieved by each recalibration method on CIFAR-100 with a perturbation severity of 1 (the lowest perturbation level). These averages are computed over all perturbations, numbers of private data sources, numbers of samples per source, and $\epsilon$ settings from the suite of experiments. Comparing these results to those shown in Table 1, which used a perturbation severity of 5 (the highest level), we see that the overall calibration improves for all methods when the degree of domain shift is lower, but our proposed algorithm, Acc-T, still outperforms other methods.

| Expected Calibration Error (median / mean) | |
| --- | --- |
| Recalibration method | CIFAR-100 |
| No recalibration | 0.1036 / 0.1342 |
| One source | 0.1067 / 0.1082 |
| Histogram binning | 0.1536 / 0.1570 |
| ECE-T | 0.1656 / 0.1680 |
| NLL-T | 0.1419 / 0.1576 |
| Acc-T | **0.0766 / 0.0841** |

Table 7: Median and mean expected calibration error (ECE) achieved for domain-shifted data with a perturbation severity of 1 under differential privacy.

