# OpenReview forum: "Privacy Preserving Recalibration under Domain Shift"
_ICLR.cc/2021/Conference — Reject_

### Official Review · AnonReviewer2 · 2020-10-27

**Rating:** 5
**Confidence:** 4

**Review:**

Summary:


The paper studies the problem of classifier recalibration under differential privacy constraints. They propose a framework with a calibrator and several private data sources, and it works as follows. At each iteration, the calibrator queries each source, and the data source sends back the private answer, which will be used to optimize the calibration. They also provide a recalibration technique, accuracy temperature scaling, which is effective under the privacy constraint for the reason of low sensitivity. Rigorous experimental results are provided.


Reasons for score:

 Overall, I am positive about this paper, but I have a few concerns.  I've listed the strengths and weaknesses below. Hopefully, the authors can address my concern in the rebuttal period. I'd be happy to raise my score if I am wrong.

Strengths:


1. The problem is well-motivated, giving the rising privacy concern and the importance of recalibration.

2. The choice of the query function is novel for privacy constraint, as it has lower sensitivity compared to the log-likelihood function.

3. They provide extensive experimental results to demonstrate the effectiveness of the proposed method.


 Weaknesses:

1. I don't see how the algorithm addresses the domain shift problem. But they claim that ''We also fine-tune on the target domain'' in section 2.2.

2. According to Tables 3 and 4 in the appendix, the Acc-T works well compared to others without privacy constraints. As it is expected to have higher biases, I would appreciate it if the authors could provide more details on how they evaluated the error and explanations.

3. The framework seems like federated learning with differential privacy, where a central server only gets private local updates from users and takes the average to optimize the parameters. The framework doesn't seem to be novel, but it can be a novel use of this setting for recalibration. It would be good to add a few sentences discussing the connections to federated learning.

4.. Truncating the log likelihood function can also lower the sensitivity. It would be interesting to see the comparison.

Minor comments:

1. In algorithm 1, the query functions for all d data sources are the same. But the general framework states that they can be different. I just wonder how to design a customized query function for each data source. Or maybe it would be more clear to remove the subscript.

---

> ### Author Response · Authors · 2020-11-23
> **Response to AnonReviewer2**
>
> We are grateful for the detailed review and thank the reviewer for their constructive comments.
>
> *“I don't see how the algorithm addresses the domain shift problem.”*
>
> We address the domain shift point in the general response above.
>
> *“According to Tables 3 and 4 in the appendix, the Acc-T works well compared to others without privacy constraints. As it is expected to have higher biases, I would appreciate it if the authors could provide more details on how they evaluated the error and explanations.”*
>
> Acc-T has **slightly higher bias but much lower variance** than the other methods, which leads to better performance unless the recalibration dataset is very large. We note that a similar observation has been made before for temperature scaling compared to histogram binning in [1]. [1] showed that for typical recalibration datasets, temperature scaling achieves lower calibration error than histogram binning, even though histogram binning is unbiased and guaranteed to be perfectly calibrated in the limit of infinite data.
>
> For evaluating the ECE error, we use exactly the same approximation method as [1], with 15 equally spaced bins. Therefore, the results should be comparable and standardized.
>
> [1] On Calibration of Modern Neural Networks, Guo et. al.
>
> *“The framework seems like federated learning with differential privacy, where a central server only gets private local updates from users and takes the average to optimize the parameters. The framework doesn't seem to be novel, but it can be a novel use of this setting for recalibration. It would be good to add a few sentences discussing the connections to federated learning.”*
>
> Good point, this is a valuable connection to make. Our problem setup can be framed as differentially private federated learning for recalibration. We have added this context to our problem statement.
>
> *“Truncating the log likelihood function can also lower the sensitivity. It would be interesting to see the comparison.”*
>
> Thank you for pointing this out! In the original paper, we used  source domain data to pick a truncation threshold for the NLL-T method. It’s true that this threshold may not result in optimal recalibration, but it does guarantee differential privacy since it does not access private data from the target domain. We also experimented with using private target domain data  to find the optimal truncation thresholds. This artificially improves the result of NLL-T by violating differential privacy. However, even with the most favorable threshold, NLL-T performs significantly worse than Acc-T. In our experiments, using the optimal threshold for NLL-T never improves its performance by more than 10% over the reported results, and this improvement comes at the cost of privacy violations. We have added an additional plot (Figure 52) in Appendix E.3, showing the ECE achieved by NLL-T at different clipping thresholds for an exemplar dataset.
>
> *“Minor comment: In algorithm 1, the query functions for all d data sources are the same. But the general framework states that they can be different. I just wonder how to design a customized query function for each data source. Or maybe it would be more clear to remove the subscript.”*
>
> Thank you for pointing this out. We have removed the subscript.

---

### Official Review · AnonReviewer1 · 2020-10-27
**Neat use of accuracy temperature scaling for differential privacy**

**Rating:** 7
**Confidence:** 4

**Review:**

The paper tackles the problem of privacy preserving calibration under domain shift, which is an interesting combination of 3 separate problems that may often occur together. The main contribution is the use of accuracy temperature scaling for calibration, which is a good match for differential privacy. Extensive experimental validation is given.

Strengths:
- The use of accuracy temperature scaling is neat in combination with DP. Whilst not a startling novelty, it is a nice observation that their method Accuracy Temperature scaling (Acc-T) combines so well with differential privacy, and yet does not lose out in terms of utility (and in fact does better under more stringent privacy settings, since fewer DP noise iterations are required.
- Strong experimental section. Although only on image data, the experiments cover a multitude of shift types, and the baselines are apt for the task.

Weaknesses:
- In terms of methodology, the paper is only tangentially related to domain shift, since there is a form of fine-tuning on the target dataset (using the overall accuracy) (see §2.2). There are extensive experiments on recalibration under domain shift (using perturbed image datasets). However, there must also be an interplay between the degree of shift, the effectiveness of transfer learning (e.g. fine-tuning) and calibration. This is not really covered

---

> ### Author Response · Authors · 2020-11-23
> **Response to AnonReviewer1**
>
> We thank the reviewer for their thoughtful feedback and positive comments.
>
> *“In terms of methodology, the paper is only tangentially related to domain shift, since there is a form of fine-tuning on the target dataset (using the overall accuracy) (see §2.2).”*
>
> We address the domain shift point in the general response above.
>
> *“However, there must also be an interplay between the degree of shift, the effectiveness of transfer learning (e.g. fine-tuning) and calibration.”*
>
> Yes, there is an interplay between the severity of the domain shift, the effectiveness of fine-tuning, and calibration. We have performed additional experiments with different domain shift severity levels (where higher severity → bigger distribution shift) for CIFAR-100. The results are shown in Table 7 of Appendix E.3. To give a high level overview of our experimental results: 1. Higher severity corruption generally leads to worse calibration performance under differential privacy. 2. Even at the lowest severity level, Acc-T achieves a median ECE of 0.0766, a 26% lower calibration error compared to the second best baseline across the board.

---

### Official Review · AnonReviewer4 · 2020-10-28
**The empirical results seem complete. I have several concerns about the technical part.**

**Rating:** 5
**Confidence:** 4

**Review:**

This paper studies the problem of privacy-preserving calibration under the domain shift. The authors propose ''accuracy temperature scaling'' with privacy guarantees.

The empirical results seem complete. I still have several concerns about the technical part.


1. The proposed algorithm is not described very clearly in section 3 and section 4. After spending considerable time reading sections 3 and 4, I still feel hard to follow how they address the domain-shift issues.

2. The privacy part seems like a plug-and-play of the Laplace mechanism. Hence, the technical novelty might be limited. Note that the privacy computation in section 3 based on a naive composition --- ` each $M_i$ satisfies $\epsilon/k$, the total privacy cost follows $\epsilon$. I would suggest the authors use recently advanced composition [1,2] for better privacy and utility tradeoffs. Moreover, the calculation of sensitivity seems to be wrong. As the authors claim in Section C.2.1 in the appendix, the sensitivity is technically infinite. They set $\triangle_f=10$ based on empirical observation, which violates the privacy definition.

[1] The Composition Theorem for Differential Privacy.
[2] Renyi Differential Privacy.

---

> ### Author Response · Authors · 2020-11-23
> **Response to AnonReviewer4**
>
> We appreciate the constructive comments and thank the reviewer for their feedback.
>
> *“The proposed algorithm is not described very clearly in section 3 and section 4.”*
>
> We have rewritten parts of Sections 3 and 4, and we will also add a new figure to improve clarity. We highlight a few new writing changes:
>
> We have added an introduction to Section 3 that clarifies how our problem setup **falls within the context of federated learning**. Multiple parties experience the same domain shift (e.g. they live in the same changing world). Each party would benefit from access to additional data, but each party also wants to keep their own data private. We propose an algorithm that allows all parties to react to domain shifts more quickly by pooling their data (so each individual party needs less labeled data from the new distribution), while maintaining the privacy of each party.
>
> To clarify the examples in Section 3.1, let us consider Example 1 in the context of federated learning. In this case, the hospitals are the parties that wish to keep their data (patient info) private. The novel strain of the virus represents a domain shift. The hospitals each have only a few data points (images of patients with the novel virus), so they want to aggregate their data in order to improve their classifier’s calibration while still respecting patient privacy.
>
> Our proposed algorithm is in Section 4.1 of the paper. On Line 2 we select initial temperature values (the recalibration parameter).  Line 3 specifies a query function that the hospitals use to pool their data while respecting differential privacy. Lines 4-12 implement differentially private golden section search over the recalibration temperature parameter. The algorithm outputs a temperature value that improves the classifier’s calibration on the new domain.
>
> *“I still feel hard to follow how they address the domain-shift issues.”*
>
> We address the domain shift point in the general response above.
>
> *“The privacy part seems like a plug-and-play of the Laplace mechanism.”*
>
> We would like to highlight that our goal is to propose the problem setup, a general framework for addressing the problem, and show the surprising empirical effectiveness of one novel algorithm (accuracy temperature scaling). Because this is the first time that this problem setup has been proposed, we use the Laplace mechanism, which works to preserve privacy. We agree that results with more advanced privacy-preserving techniques are an interesting area for future work!
>
> *“I would suggest the authors use recently advanced composition [1,2] for better privacy and utility tradeoffs.”*
>
> We use pure differential privacy as our notion of privacy. These advanced compositions are applicable to more relaxed definitions of privacy. In principle our framework should also work in a plug-and-play manner with these relaxed definitions of privacy and advanced compositions, and we leave that for future work.
>
> *“As the authors claim in Section C.2.1 in the appendix, the sensitivity is technically infinite.”*
>
> We use a clipping threshold to preserve privacy when the sensitivity is infinite. Please see “Bounding the sensitivity of $\Delta f$ for NLL-T” in the general response above.

---

### Official Review · AnonReviewer3 · 2020-10-29
**This paper studies the problem of recalibrating a classifier under the presence of domain shift and the constraints of differential privacy.  They show how to adapt several algorithms for dealing with domain shift to the paradigm of differential privacy, giving mechanisms that achieve both goals.**

**Rating:** 6
**Confidence:** 3

**Review:**

Summary:

This paper studies the problem of recalibrating a classifier under the presence of domain shift and the constraints of differential privacy.  They show how to adapt several algorithms for dealing with domain shift to the paradigm of differential privacy, giving mechanisms that achieve both goals.

Strong Points:

1. Addresses a new, interesting, and practically relevant problem.
2. Technically strong, good insights and clearly bridges the gap between two distinct areas.
3. Well organized and written, very clear for the most part.

Weak Points:

1. The sensitivity of f cannot be bounded.  What the authors propose (Section C.2.1) to address this --- i.e., using a sufficiently large value based on the empirical values --- is not generally accepted.

Other Notes:

I think the ideas are cool.  The reduction to a 1-dimensional minimization problem over T makes sense.  The technical insight to use golden search to reduce queries/noise is clever.  Experiments show that the proposed approach actually works as it is expected to.  My one weak point is a bit concerning however.  Hopefully authors can address it adequately in the rebuttal.

---

> ### Author Response · Authors · 2020-11-23
> **Response to AnonReviewer3**
>
> We thank the reviewer for their positive feedback and constructive comments.
>
> *“The sensitivity of f cannot be bounded. What the authors propose (Section C.2.1) to address this --- i.e., using a sufficiently large value based on the empirical values --- is not generally accepted.”*
>
> We use a clipping threshold to preserve privacy when the loss function is unbounded. Please see "Bounding the sensitivity of $\Delta f$ for NLL-T" in the general response above.

---

### Author Response · Authors · 2020-11-23
**General Response**

We thank all reviewers for their thoughtful feedback on our work. We would like to clarify a few points below:

**Bounding the sensitivity of $\Delta f$ for NLL-T:**

Our framework is guaranteed to preserve privacy even when the loss function is unbounded (as in the case of NLL-T). During deployment, if the loss function is unbounded, a clipping threshold should be selected ***before*** the (private) recalibration data is observed, i.e. loss_clipped = min(clipping_threshold, loss). The Laplace mechanism then guarantees differential privacy, using standard analysis. In our experiments, we selected good clipping thresholds based on the public training data (rather than the private recalibration data) for the NLL-T baseline. Note that this discussion only applies to the NLL-T baseline; our recommended algorithm, Acc-T, does not need any clipping.

**Relationship to domain shift:**

Our framework should not be confused with unsupervised domain adaptation. Instead, our recalibration framework requires a small amount of labeled data from the target domain. Specifically, we take as input a classification model trained on data from some source domain. We then fine-tune the model using a small amount of labeled data from the target domain while preserving differential privacy. Our framework returns a model with improved calibration on the target domain.

Our method (Acc-T) is indeed **applicable outside the context of domain shift** (e.g. it improves calibration on the source domain as well, see the experiments in Appendix E.3). However, it is particularly suitable for the domain shift setup because our method assumes access to very little information about the new domain (simply the average overall accuracy). Thus, **very little target domain data is needed** to estimate the average accuracy, and it is easy to preserve privacy.

---

### Decision · Program_Chairs · 2021-01-07
**Final Decision**

**Decision:**

Reject

**Comment:**

This work considers the problem of calibrating a multi-class classifier while preserving differential privacy. It proposes a method Accuracy Temperature Scaling, that aims to achieve consistency rather than calibration. The method is particularly easy to implement under the constraint of DP. The paper then evaluates  the calibration algorithm in the context of domain perturbation/shift and, as the authors demonstrate it outperforms adaptations of other technques to DP.

The strong sides of this work are
* the first work to study calibration in this setting (albeit that is also a result of the setting being of a relatively narrow interest)
* proposes a new algorithm
* evaluation on multiple benchmarks

The weaknesses
* The method is not justified either by theoretical analysis or clear intuition
* Evaluation of performance in the context of domain shift makes the the presentation somewhat confusing and experiments much more involved but is largely orthogonal to the problem of calibration

Overall the work has merits but also significant issues.